# Nicotine exacerbates MASH via inducing intestinal dysbiosis and barrier dysfunction
Fangfang Yi [1], Jinyong Wang [1], Yi Chen [1], Yuanhang Xia [2], Hongfei Zhou [1], Yilun Huang [3], Yujuan Shen[1], Sizhe Fang [1], Xiaodong Wang[1], Ya Zhang[1], Yongping Chen [1] ✉ & Dazhi Chen [2] ✉

Nicotine accumulation in the intestine is associated with an exacerbation of metabolic dysfunction-associated steatohepatitis (MASH), but the underlying mechanisms remain enigmatic. We investigated how nicotine impacted intestinal microbiota composition and barrier function in MASH. Our study revealed significant intestinal microbiota dysbiosis and upregulated hypoxia-inducible factor 1-alpha (HIF-1α) levels in nicotine-exposed MASH mice. HIF-1α knockdown worsened intestinal barrier dysfunction in nicotine-exposed MASH mice. This exacerbation resulted from the suppression of MEK/ERK signaling pathway phosphorylation in *HIF-1α*-deficient mice. *Lactobacillus rhamnosus* GG supernatant can alleviate hepatic injury in nicotine-exposed MASH mice; however, this protective effect was abolished in the absence of *HIF-1α*. Taken together, this study reveals a critical pathologic role of nicotine in exacerbating MASH through intestinal microbiota disruption and barrier dysfunction, which is associated with the downregulation of HIF-1α in the intestine. It also suggests exogenous probiotic supplementation as a potential therapeutic strategy for mitigating nicotine-induced MASH progression.

Metabolic dysfunction-associated steatotic liver disease (MASLD) has emerged as the most common chronic liver disease worldwide, affecting approximately 30% of adults globally. Alarmingly, the global prevalence of MASLD is estimated to rise to 55% by 2040[1]. MASLD, the fastest-growing cause of hepatocellular carcinoma, has an unclear pathogenesis. However, evidence links it to risk factors such as smoking, unhealthy diets, insulin resistance, type 2 diabetes mellitus, increased hepatic lipogenesis, and intestinal microbiota dysbiosis[2].

Smoking, as a well-recognized detrimental lifestyle habit, is known to increase the risk of lung and colorectal cancers[3]. Nicotine, the primary addictive component of tobacco, has been identified as a key factor in smoking-related health issues[4]. Our previous study demonstrated that nicotine induced mitochondrial dysfunction, oxidative stress, and apoptosis through suppression of CDGSH iron sulfur domain 3 expression, thereby exacerbating MASLD[5].

Intestinal dysbiosis and barrier dysfunction disrupt intestinal homeostasis, exacerbate systemic inflammation, and promote the translocation of harmful substances, further aggravating MASLD[6]. Evidence indicates that nicotine accumulates in the intestine, exacerbating metabolic dysfunction-

associated steatohepatitis (MASH). However, supplementation with intestinal microbiota has been shown to degrade intestinal nicotine and alleviate smoking-induced MASH[7]. *Lactobacillus rhamnosus* GG is a widely studied probiotic that regulates the intestinal microbiota, improves lipid metabolism and inflammation, and effectively alleviates liver damage and metabolic disorders in high-fat diet-induced MASLD mice[8]. Moreover, *Lactobacillus rhamnosus* GG enhances hypoxia-inducible factor 2α signaling and upregulates the expression of intestinal tight junction proteins, thereby preserving intestinal barrier integrity and alleviating alcoholic liver injury[9]. Hypoxia-inducible factor -1alpha (HIF-1α) plays a pivotal role in regulating intestinal homeostasis. It modulates intestinal epithelial integrity, and its dysregulation is associated with increased intestinal permeability and inflammatory signaling[10,11]. Our previous study demonstrated that mice with intestinal epithelial-specific knockout of HIF-1α exhibited more severe liver injury and higher serum lipopolysaccharides (LPS) levels in a murine model of alcoholic liver disease, further confirming the critical role of HIF-1α in maintaining intestinal microbiota homeostasis and gut barrier integrity[10]. However, the role of HIF-1α in nicotine-exposed MASH remains inadequately understood.

[1]Hepatology Diagnosis and Treatment Center, The First Affiliated Hospital of Wenzhou Medical University & Zhejiang Provincial Key Laboratory for Accurate Diagnosis and Treatment of Chronic Liver Diseases, Wenzhou, Zhejiang, China. [2]School of Clinical Medicine, The First People's Hospital of Lin'an Distract, Hangzhou, Lin'an People's Hospital Affiliated to Hangzhou Medical College, Hangzhou Medical College, Hangzhou, Zhejiang, China. [3]Alberta Institute, Wenzhou Medical University, Wenzhou, China. ✉e-mail: cyp@wmu.edu.cn; dazhichen@126.com

In this study, we investigated how nicotine affects intestinal microbiota dysbiosis, barrier dysfunction, and HIF-1α expression, elucidating the mechanisms by which nicotine exacerbates MASH. By providing new insights into the gut-liver axis, our findings enhance the understanding of smoking-related MASH pathogenesis and identify potential therapeutic targets.

## Results

### Nicotine exposure aggravates liver tissue damage in MASH mice

To study the effect of nicotine on MASH, we intraperitoneally injected normal chow and CDAHFD-fed mice with saline or nicotine (1.5 mg/kg body weight) for 10 weeks (Fig. 1a). Nicotine had no significant effect on hepatic steatosis and fibrosis in mice fed a normal diet, but nicotine

**Fig. 1 | Nicotine exposure aggravates liver tissue damage in MASH mice. a** Animal model flow chart. **b** H&E (upper), Masson's trichrome (middle), and Oil Red O (lower) staining of the liver in mice. Scale bar = 200 μm. **c** Serum alanine aminotransferase (ALT) and aspartate aminotransferase (AST) in mice. **d** Hepatic triglyceride (TG) content. **e** Hepatic total cholesterol (TCHO) content. **f–i** Hepatic gene expression of pro-inflammatory cytokines *Il1b*, *Il6* and *Tnf-α*, and anti-inflammatory cytokines *Il10*. **j** CD20 staining of the liver. Scale bar = 200 μm. **k** F4/80 staining of the liver. Scale bar = 200 μm. **l** Ly6G staining of the liver. Scale bar = 200 μm. $^{*}P < 0.05$, $^{**}P < 0.01$, $^{***}P < 0.001$, $^{****}P < 0.0001$, ns: not significant. Data are presented as mean ± SD; *n* = 6.

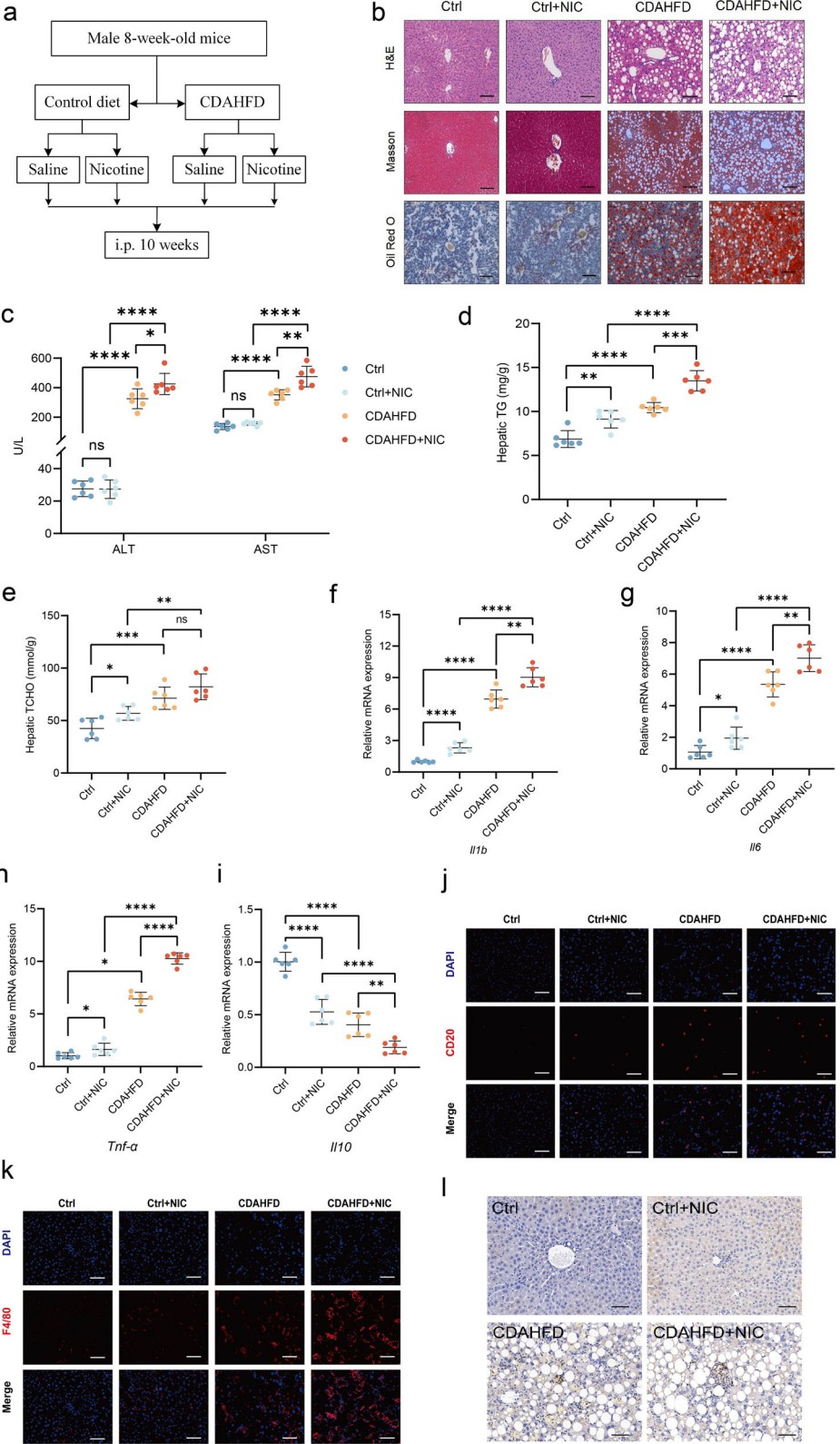

significantly increased hepatic steatosis and fibrosis under the CDAHFD diet (Fig. 1b). Serum ALT and AST levels, as well as triglyceride and cholesterol levels in liver tissue, were significantly higher in nicotine-treated mice compared to the control group (Fig. 1c–e). Additionally, pro-inflammatory cytokines Il6, Il1b, and Tnf-α were significantly elevated in the liver of nicotine-treated mice, while the anti-inflammatory cytokine Il10 was significantly decreased (Fig. 1f–i). CD20 and F4/80 immunofluorescence staining and Ly6G immunohistochemical staining showed increased infiltration of B lymphocytes (Fig. 1j), macrophages (Fig. 1k), and neutrophils (Fig. 1l), respectively, in nicotine-treated CDAHFD mice. As illustrated in Supplementary Fig. 1, RNA sequencing analysis of the liver revealed significant changes in genes closely associated with lipid metabolism in nicotine-exposed CDAHFD mice compared to those receiving only CDAHFD, such as Cyp2c70, Lpin1, Elovl5. These results suggest that nicotine exacerbated liver inflammation in CDAHFD mice.

### Nicotine alters the intestinal microbiota composition in MASH mice

To investigate whether the effects of nicotine on MASH were related to intestinal tissue damage and alterations in the intestinal microbiota, we first analyzed the intestinal tissues of mice in each group. H&E staining revealed that, compared to other groups, nicotine-exposed CDAHFD mice exhibited disordered intestinal epithelial structure and loss of epithelial cells (Fig. 2a). Then, we analyzed the fecal microbiota by performing 16S sequencing of the fecal samples. The ASVs among the four groups were compared, including 588 ASVs in the control group, 415 in the CDAHFD group, 777 in the nicotine group, and 384 cases in the CDAHFD group with nicotine exposure, with 147 of those existing in all groups (Fig. 2b). Nicotine-exposed CDAHFD group had significantly reduced alpha diversity compared to CDAHFD alone group (Fig. 2c, d). PCoA analysis (beta diversity) showed that the clustering of the intestinal microbiota of the four groups of mice was significantly different due to different diets (Fig. 2e). At the phylum level, Bacteroidota was significantly decreased, whereas Verrucomicrobiota was significantly increased in the nicotine-exposed CDAHFD group (Fig. 2f). At the genus level, Akkermansia and Erysipelatoclostridium were significantly increased in nicotine-exposed CDAHFD group (Fig. 2g). In addition, LEfSe was used to compare the species abundance differences between CDAHFD and nicotine-exposed CDAHFD group. We found that Akkermansia and Erysipelatoclostridium were significantly increased in nicotine-exposed CDAHFD group (Fig. 2h). These results suggest that nicotine exposure may affect the intestinal microbial composition of these specific bacterial families, leading to intestinal microecological balance and thus aggravating MASH.

### Nicotine impairs intestinal barrier function in MASH mice

Increased bacterial translocation may be related to impaired intestinal barrier function. To study the effects of nicotine on intestinal barrier function, the expression levels of intestinal tight junction proteins ZO-1, occludin, claudin-1, and serum LPS levels were analyzed. PCR, western blot, and immunofluorescence staining showed the expressions of ZO-1, occludin, claudin-1 were significantly decreased in the nicotine-exposed CDAHFD group (Fig. 3a–j). The mucus secreted by goblet cells plays a critical role in maintaining mucosal barrier function. Nicotine exposure reduced the number of goblet cells in the intestinal mucosa, with a more pronounced reduction observed in the CDAHFD+NIC group (Supplementary Fig. 2). We further confirmed the damaged tight junctions by TEM (Fig. 3k). Meanwhile, we found that the intestinal pro-inflammatory factors, serum LPS levels, and abundance of Erysipelatoclostridiaceae were significantly increased in the nicotine-exposed CDAHFD group compared with the other groups (Fig. 3l–o). F4/80 and CD3 immunofluorescence staining showed increased infiltration of T lymphocytes and macrophages in nicotine-exposed CDAHFD mice (Supplementary Fig. 3).

In addition, RNA sequencing analysis was performed on intestinal tissue samples to determine potential alterations in the intestinal genes of CDAHFD mice induced by nicotine exposure. Compared with CDAHFD group,

nicotine-exposed CDAHFD group had a total of 305 genes with significant differential expression, of which 227 genes were upregulated and 78 genes were downregulated (Fig. 4a). In the nicotine-exposed CDAHFD group, the expressions of antimicrobial peptides, including Defa2, Defa3, Defa5, Reg3b, Reg3d, Reg3g, Ang4, and Lyz1, were significantly reduced, indicating a severe impairment of intestinal immune barrier function (Fig. 4b). The DEGs between the CDAHFD and nicotine-exposed CDAHFD groups were analyzed based on GO. Defense response to bacteria in the biological process category, immunoglobulin receptor binding in the molecular function category, extracellular region in the cellular component category were the most enriched (Fig. 4c). These results collectively indicate that nicotine exacerbates intestinal barrier dysfunction in MASH mice.

### Antibiotic treatment mitigates nicotine-induced liver damage in MASH mice

To further analyze whether nicotine-induced liver damage was mediated by intestinal microbiota dysbiosis, the mice were treated with an Abx (Fig. 5a). First, 16S rRNA sequencing was performed on fecal samples, confirming effective clearance of intestinal flora, as evidenced by significantly reduced α diversity and distinct β diversity between groups (Supplementary Fig. 4a–e). Second, H&E staining revealed that nicotine-exposed CDAHFD mice exhibited reduced liver and intestinal injury following Abx treatment (Fig. 5b). The serum ALT, AST, and LPS levels, as well as triglyceride and cholesterol levels in liver tissues, were significantly reduced in the nicotine-exposed CDAHFD group after Abx treatment (Fig. 5c-f). In the nicotine-exposed CDAHFD group, intestinal pro-inflammatory factors were significantly reduced after Abx treatment (Fig. 5g–i). PCR and western blot demonstrated that the expression levels of ZO-1, occludin, and claudin-1 were markedly upregulated in the nicotine-exposed CDAHFD group after Abx treatment (Fig. 5j–p). Additionally, RNA sequencing analysis of intestinal tissues showed that antimicrobial peptides, including Defa2, Reg3g, and Lyz1, were markedly restored in the nicotine-exposed CDAHFD group following Abx treatment (Fig. 5q). Finally, at the genus level, significant reductions in Akkermansia and Erysipelatoclostridium were observed after Abx treatment (Supplementary Fig. 4f). These findings suggest that intestinal microbiota dysbiosis mediates nicotine-induced liver injury in MASH mice.

### HIF-1α knockdown exacerbates nicotine-exposed MASH in mice

Study has demonstrated that HIF-1α plays a key role in maintaining intestinal barrier integrity[12]. Western blot analysis showed a marked upregulation of HIF-1α expression in nicotine-exposed MASH mice (Fig. 6a, b). We speculate that nicotine exposure may exacerbate the intestinal anaerobic environment in MASH mice, which could contribute to the upregulation of HIF-1α. To test this, we administered Abx to deplete the intestinal microbiota. Nuclear protein fractions were subsequently extracted and subjected to Western blot analysis. The results showed that nuclear HIF-1α levels were significantly reduced after Abx treatment, supporting our hypothesis and indicating that the transcriptional activation of HIF-1α was attenuated under microbiota-depleted conditions (Fig. 6c, d). To further explore the mechanistic role of HIF-1α, we used tail vein injection of AAV7-HIF-1α to effectively knock down HIF-1α expression (Fig. 6e). As shown in Fig. 6f–h, AAV7-HIF-1α markedly reduced HIF-1α mRNA and protein levels in the ileum, whereas no significant change was observed in the liver compared with AAV7-Vector control. These results confirm that the AAV7 vector predominantly targets ileum tissue and achieves efficient HIF-1α knockdown in the gut, while exerting minimal impact on hepatic HIF-1α expression. As shown in Fig. 6i, liver damage was exacerbated in nicotine-exposed MASH mice with HIF-1α knockdown (AAV7-HIF-1α group) relative to the AAV7-Vector group. After silencing HIF-1α expression, the levels of tight junction proteins ZO-1, occludin, and claudin-1 were significantly reduced in nicotine-exposed MASH mice (Fig. 6j–p). Pro-inflammatory cytokines (Il6, Il1b, Tnf-α) and the anti-inflammatory factor Il10 exhibited significant changes in the nicotine-exposed MASH group (Supplementary Fig. 5a–d). Additionally, we found that the phosphorylation level of MEK/ERK was

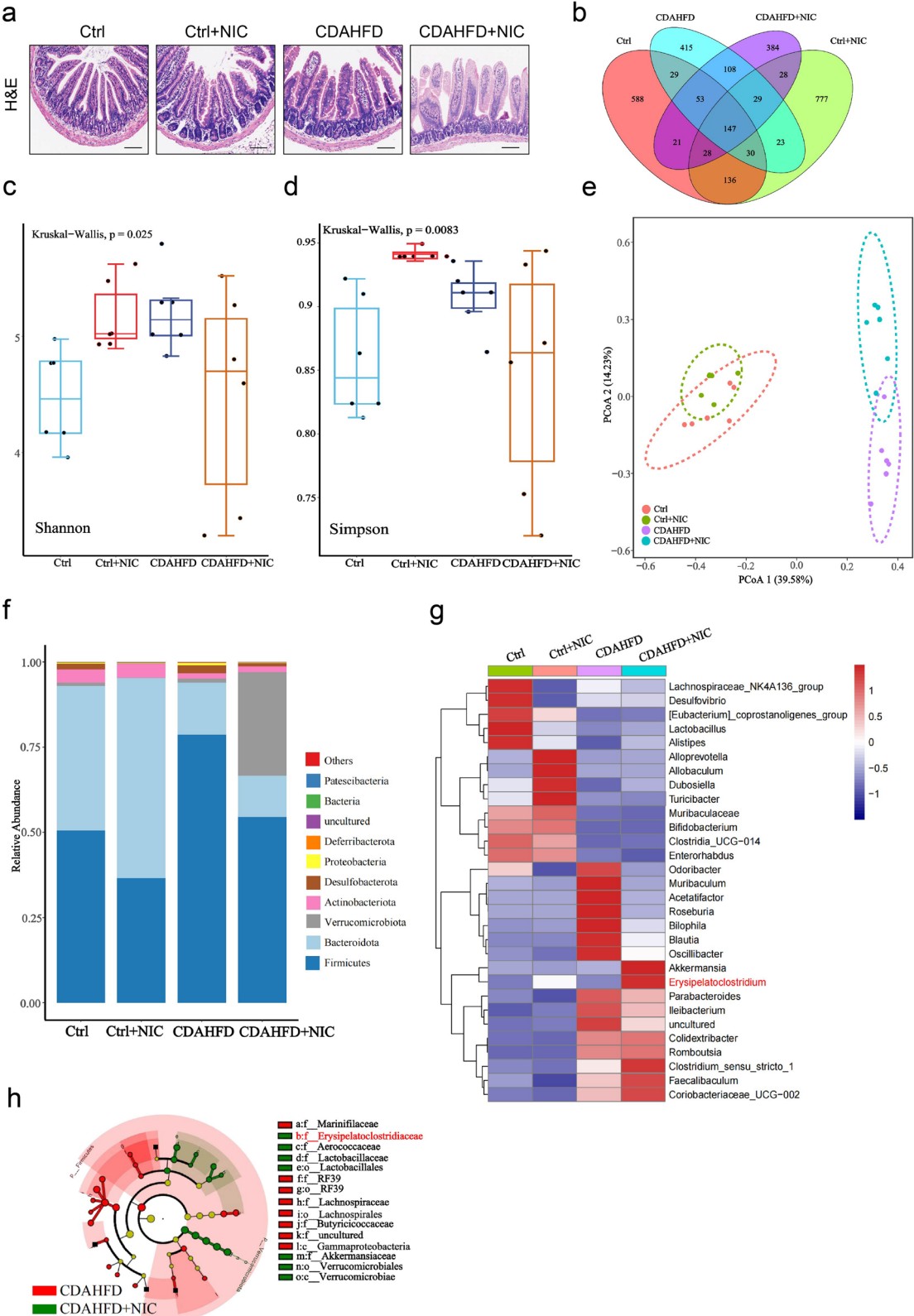

**Fig. 2 | Nicotine alters the gut microbiota composition in MASH mice. a** H&E staining of the intestine in mice. Scale bar = 200 μm. **b** Venn diagram. **c, d** alpha diversity results. **e** beta diversity with PCoA. **f** Relative abundance of bacteria at the phylum level. **g** Heatmap of bacteria at the genus level. **h** Cladogram plot of LEfSe analysis. Data are presented as mean ± SD; *n* = 6.

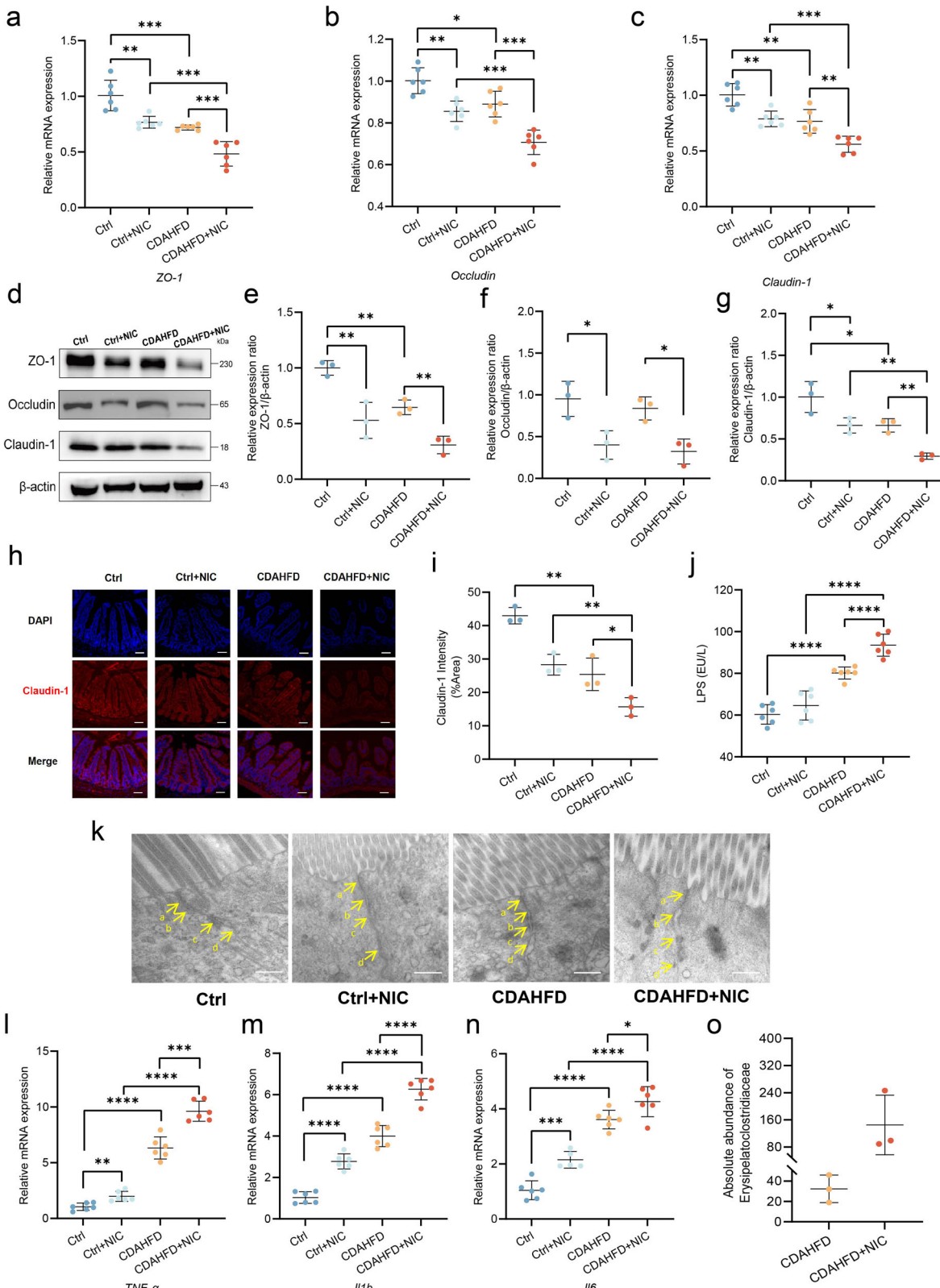

**Fig. 3 | Nicotine impairs intestinal barrier function in MASH mice. a–c** Gene expression of *ZO-1*, *occludin*, and *claudin-1* (*n* = 6). **d–g** Western blot analysis for ZO-1, occludin, and claudin-1 normalized by β-actin (*n* = 3). **h, i** IF staining and quantification of claudin-1. Scale bar = 200 μm. **j** Serum lipopolysaccharides (LPS) level (*n* = 6). **k** The representative images of the structure of the intestinal barrier of mice. Arrows point to cell-cell junction under transmission electron microscopy

(TEM). **a** Tight junction; **b** adherens junction; **c** desmosome; **d** gap junction. **l–n** Intestinal gene expression of pro-inflammatory cytokines *Tnf-α*, *Il1b*, and *Il6*, and anti-inflammatory cytokines *Il10* (*n* = 6). **o** Absolute abundance of *Erysipelatoclostridiaceae* (*n* = 3). *P* < 0.05, **P* < 0.01, ***P* < 0.001, ****P* < 0.0001. Data are presented as mean ± SD.

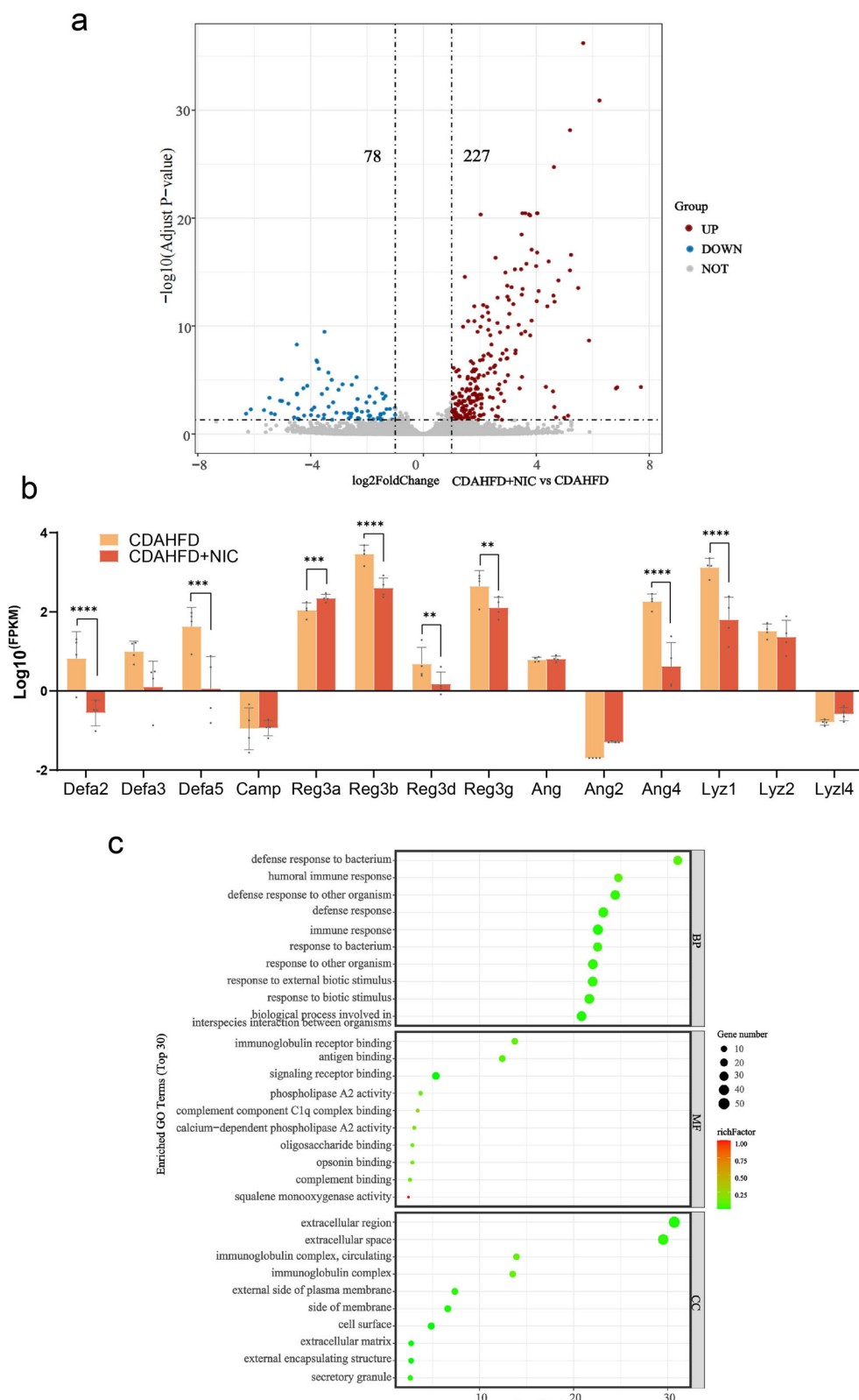

**Fig. 4 | Changes of nicotine on intestinal gene expression in MASH mice.**
**a** Volcano plot. **b** Changes in relative mRNA levels of genes related to intestinal barrier according to results of gene sequencing of intestinal tissues. **c** GO enrichment bubble map of differentially expressed genes in intestinal tissues. FPKM: fragments per kilobase of exon model per million mapped fragments. $^{**}P < 0.01$, $^{***}P < 0.001$, $^{****}P < 0.0001$. Data are presented as mean ± SD; $n = 4$.

markedly reduced in HIF-1α-deficient mice, which may underlie the more severe intestinal barrier damage observed in this group (Fig. 6q–s). These results suggest that the upregulation of HIF-1α represents an adaptive response to mitigate intestinal barrier dysfunction.

## HIF-1α is crucial for the therapeutic effect of LGG-s in nicotine-exposed MASH mice

Our previous research has demonstrated that the metabolic products of LGG-s contain a substantial amount of short-chain fatty acids, which can

**Fig. 5 | Antibiotic treatment mitigates nicotine-induced liver damage in MASH mice. a** Schematic of antibiotic cocktail (Abx). **b** H&E staining of the intestine and liver in mice. Scale bar = 200 μm. **c** Serum LPS level (*n* = 6). **d** Serum ALT and AST levels in mice (*n* = 6). **e** Hepatic TG content (*n* = 6). **f** Hepatic TCHO content (*n* = 6). **g–i** Intestinal gene expression of pro-inflammatory cytokines *Tnf-α*, *Il1b*, and *Il6* after Abx treatment (*n* = 6). **j–l** Gene expression of *ZO-1*, *occludin*, and *claudin-1* (*n* = 6). **m–p** Western blot analysis for ZO-1, occludin, and claudin-1 normalized by β-actin (*n* = 3). **q** Changes in relative mRNA levels of genes related to intestinal barrier according to results of gene sequencing of intestinal tissues. *$P < 0.05$, **$P < 0.01$, ***$P < 0.001$, ****$P < 0.0001$. Data are presented as mean ± SD.

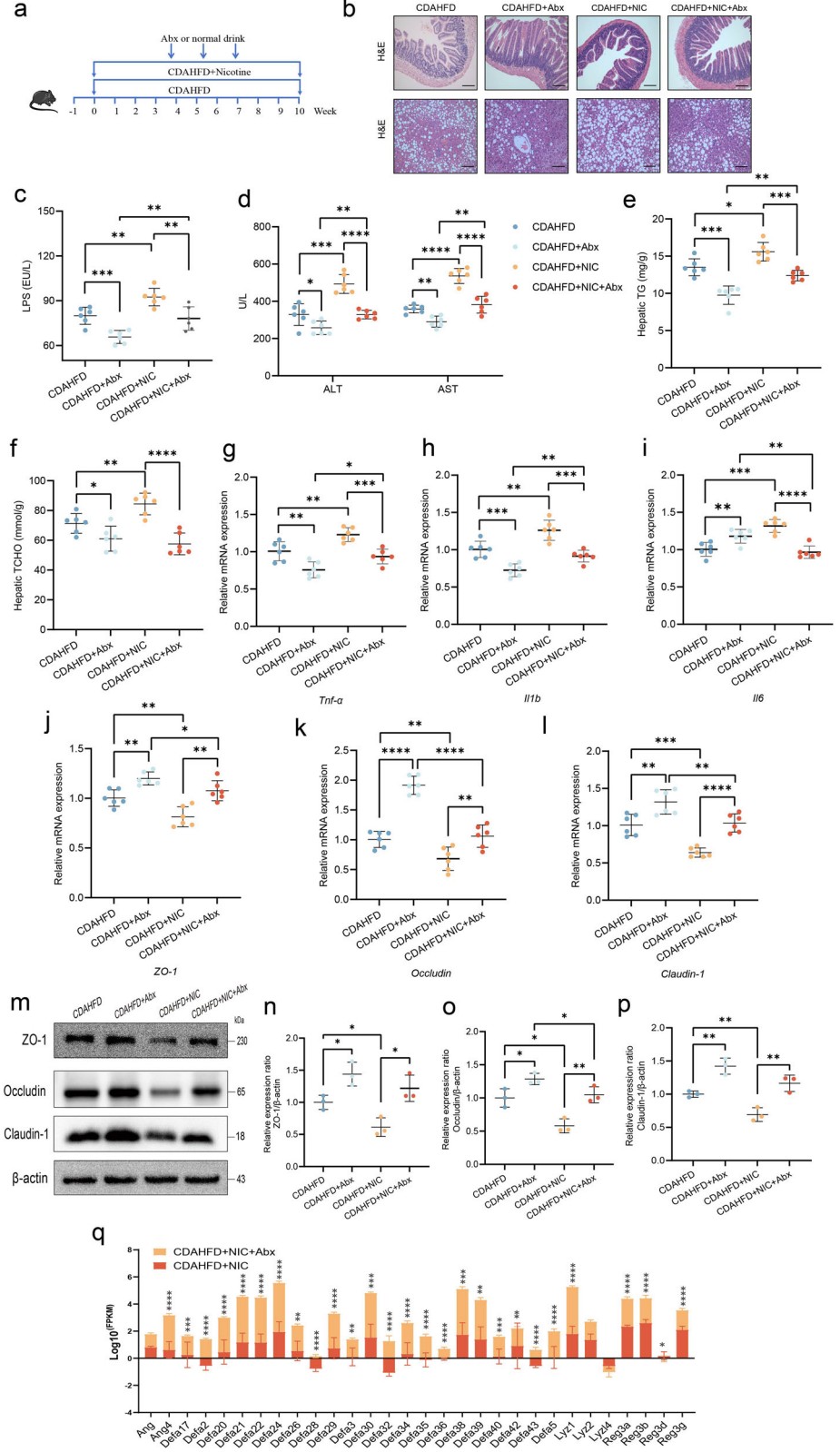

effectively improve the intestinal microbiome in autoimmune hepatitis[13]. However, the effects of LGG-s in nicotine-exposed MASH mice remain unclear. To investigate whether the therapeutic effects of LGG-s require HIF-1α, we administered daily gavage of 1 ml LGG-s (10^9/ml) or saline to nicotine-exposed MASH mice for 10 weeks. Nicotine-exposed MASH mice showed a significant improvement in serum ALT and AST levels following LGG-s treatment; however, no improvement was observed in HIF-1α

knockdown mice (Fig. 7a–c). In nicotine-exposed MASH mice, LGG-s administration in the AAV7-Vector group significantly reduced liver inflammation compared to the AAV7-*HIF-1α* group (Fig. 7d–g). Additionally, LGG-s administration upregulated the expression of mRNA and protein for intestinal tight junctions in the AAV7-Vector group mice, a finding further confirmed by TEM, whereas no such effects were observed in the AAV7-*HIF-1α* mice (Fig. 7h–o). These results demonstrated that HIF-

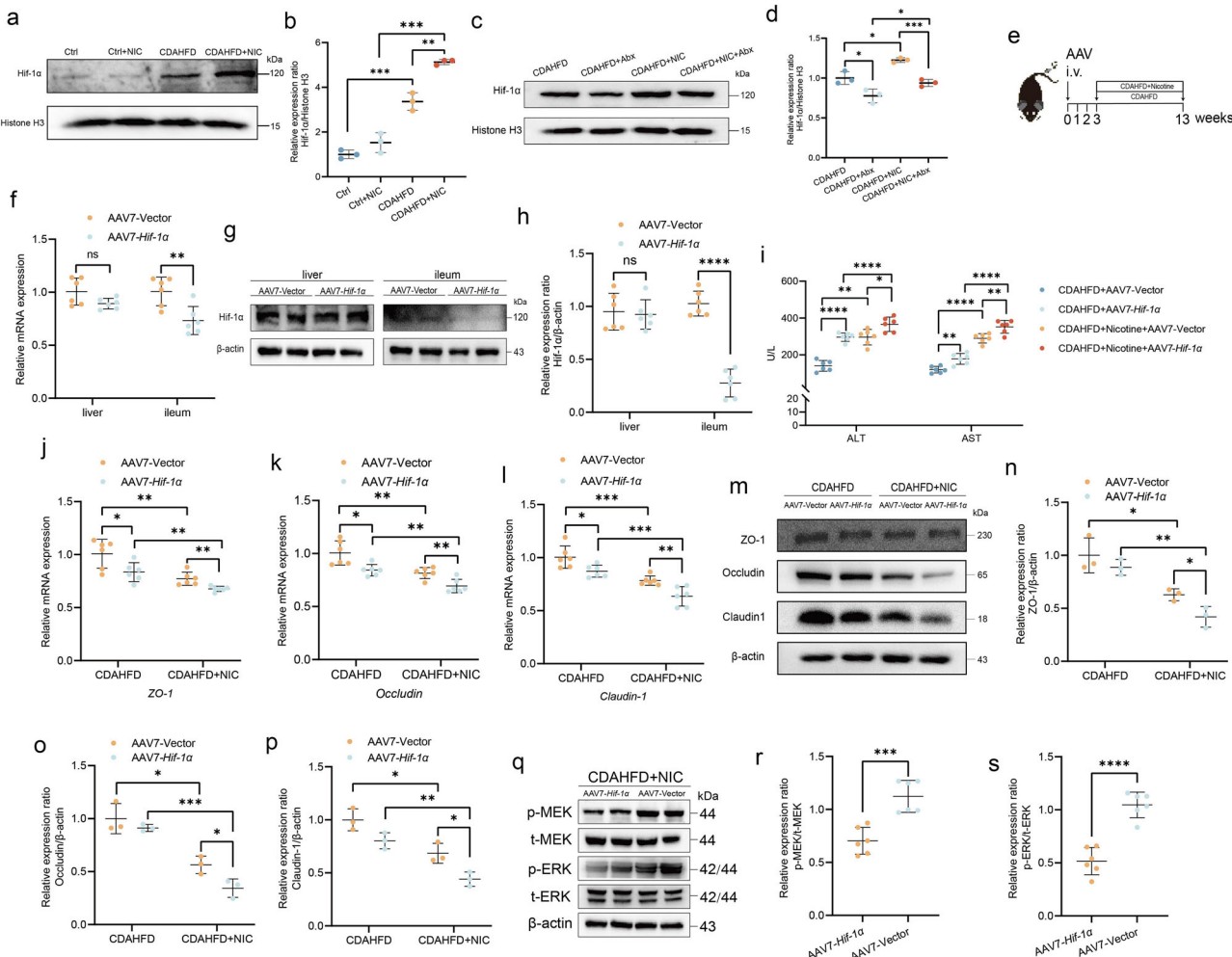

**Fig. 6 | HIF-1α knockdown exacerbates nicotine-exposed MASH in mice.**
**a, b** Western blot analysis for Hif-1α normalized by β-actin. **c, d** Western blot analysis for Hif-1α normalized by β-actin after antibiotic cocktail (Abx). **e** Schematic for the transfection of AAV7-*Hif-1α* and the establishment of the mouse nicotine-exposed MASH model (*n* = 6). **f–h** PCR and Western blot confirmed successful

knockdown of Hif-1α. **i** Serum ALT and AST levels in mice (*n* = 6). **j–l** Gene expression of *ZO-1*, *occludin*, and *claudin-1* (*n* = 6). **m–p** Western blot analysis for ZO-1, occludin, and claudin-1 normalized by β-actin (*n* = 3). **q–s** Western blot analysis for the p-MEK and p-ERK normalized by MEK and ERK (*n* = 3). $^*P < 0.05$, $^{**}P < 0.01$, $^{***}P < 0.001$, $^{****}P < 0.0001$. Data are presented as mean ± SD.

1α is essential for the therapeutic effects of LGG-s in nicotine-exposed MASH mice.

## HIF-1α/MEK/ERK pathways in LGG-s intestinal protection

*HIF-1α* was knocked down in Caco-2 cells using *HIF-1α*-specific siRNA, revealing that LGG-s failed to reverse tight junction protein damage in the nicotine-exposed *HIF-1α$^{-/-}$* group (Fig. 8a, b). This finding suggests that *HIF-1α* deficiency abolishes the protective effects of LGG-s on the intestinal barrier. Additionally, we used the MEK inhibitor U0126 to further investigate whether the therapeutic effect of LGG-s on the intestinal epithelial barrier is associated with the MEK/ERK signaling pathway. Treatment with the MEK inhibitor U0126 suppressed the LGG-s-induced upregulation of ZO-1, occludin, and claudin-1 expression, confirming that the activation of the MEK/ERK signaling pathway by LGG-s is *HIF-1α*-dependent (Fig. 8c, d).

## Discussion

The gut-liver axis plays a pivotal role in the pathogenesis of MASH. Nicotine has been shown to exacerbate MASH[7]. However, whether nicotine aggravates MASH by modulating intestinal dysbiosis and impairing intestinal barrier integrity is not yet fully understood. In this study, we found that nicotine aggravates hepatic steatosis, fibrosis, and liver injury by promoting intestinal dysbiosis and disrupting intestinal integrity.

Furthermore, the deletion of intestinal HIF-1α was found to exacerbate these liver injuries.

Nicotine is a recognized risk factor for the development of MASH[5]. However, the mechanisms by which nicotine exacerbates MASH are not yet fully understood. Although several studies have reported an association between intestinal microbiota and smoking, it remains unclear whether nicotine-induced alterations in the microbiota play a significant role in the onset and progression of MASH[3,7,14]. To address this, we established a nicotine-exposed MASH mouse model through intraperitoneal injections and conducted 16S rRNA sequencing of fecal samples. Our study revealed severe intestinal microbiota dysbiosis in nicotine-exposed MASH mice, marked by an elevated abundance of *Erysipelatoclostridium*. This finding aligns with previous studies, which showed that *Erysipelatoclostridium* was significantly enriched in mice with inflammatory bowel disease and positively correlated with TNF-α levels[15,16]. Moreover, its increased abundance has been linked to enhanced intestinal inflammation and epithelial permeability, suggesting that *Erysipelatoclostridium* may act as a potential pathogenic microorganism[17,18]. Interestingly, a notable elevation in *Akkermansia* was detected in nicotine-exposed MASH mice. While exogenous supplementation of *Akkermansia* has been shown to exert protective effects against MASH[19], the endogenous elevation of *Akkermansia* may modulate the intestinal innate immune response and facilitate bacterial translocation by degrading intestinal mucus[20]. Therefore, the causal relationship between

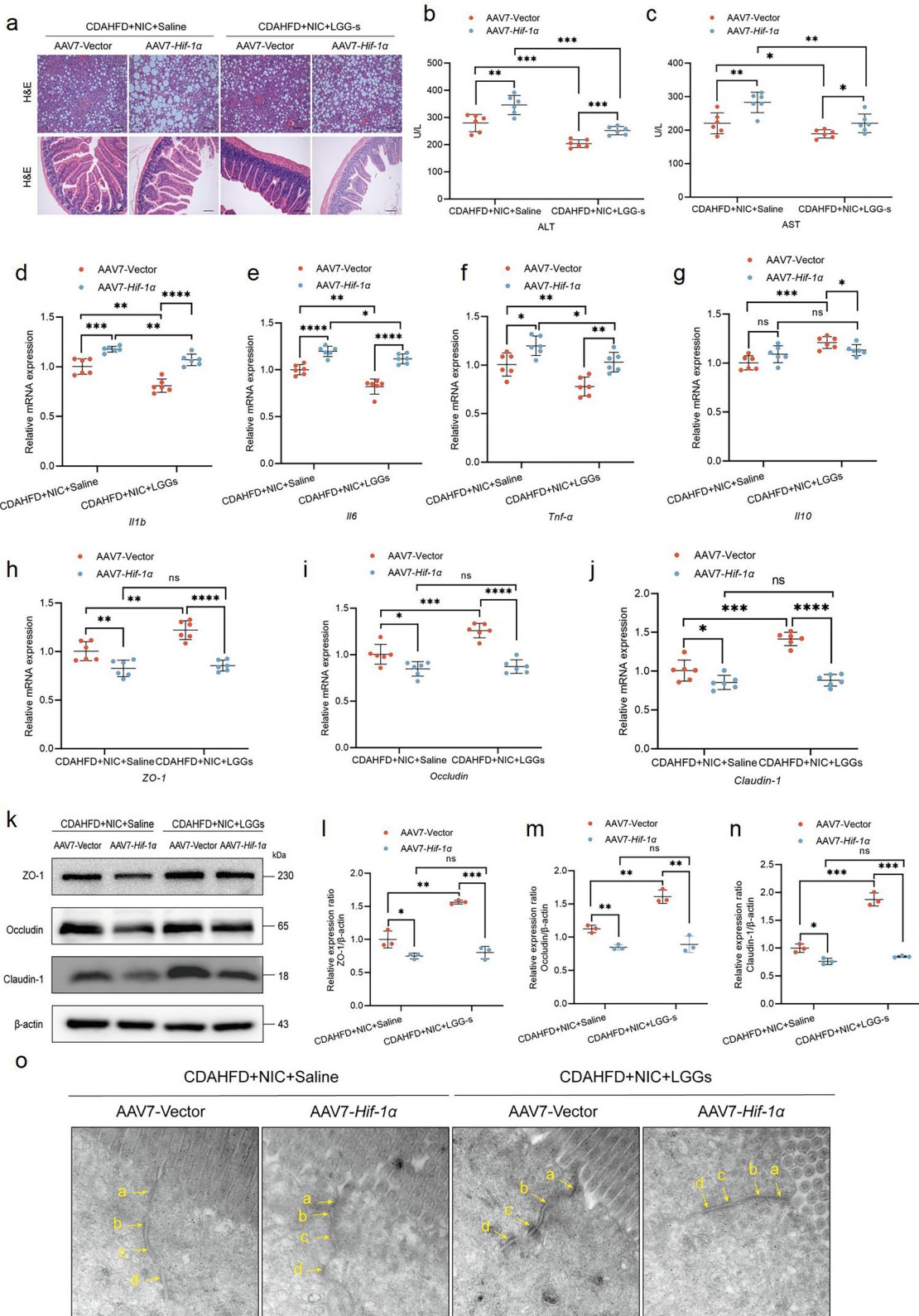

**Fig. 7 | HIF-1α is crucial for the therapeutic effect of LGG-s in nicotine-exposed MASH mice. a** H&E staining of the intestine and liver in mice. Scale bar = 200 μm. **b**, **c** Serum ALT and AST levels in mice ($n = 6$). **d–g** Intestinal gene expression of pro-inflammatory cytokines *Il1b*, *Il6*, and *Tnf-α*, and anti-inflammatory cytokines *Il10* ($n = 6$). **h–j** Gene expression of *ZO-1*, *occludin*, and *claudin-1* ($n = 6$). **k–n** Western blot analysis for ZO-1, occludin, and claudin-1 normalized by β-actin ($n = 3$). **o** The representative images of the structure of the intestinal barrier of mice. Arrows point to cell-cell junction under TEM. **a** Tight junction; **b** adherens junction; **c** desmosome; **d** gap junction. $^{*}P < 0.05$, $^{**}P < 0.01$, $^{***}P < 0.001$, $^{****}P < 0.0001$. Data are presented as mean ± SD.

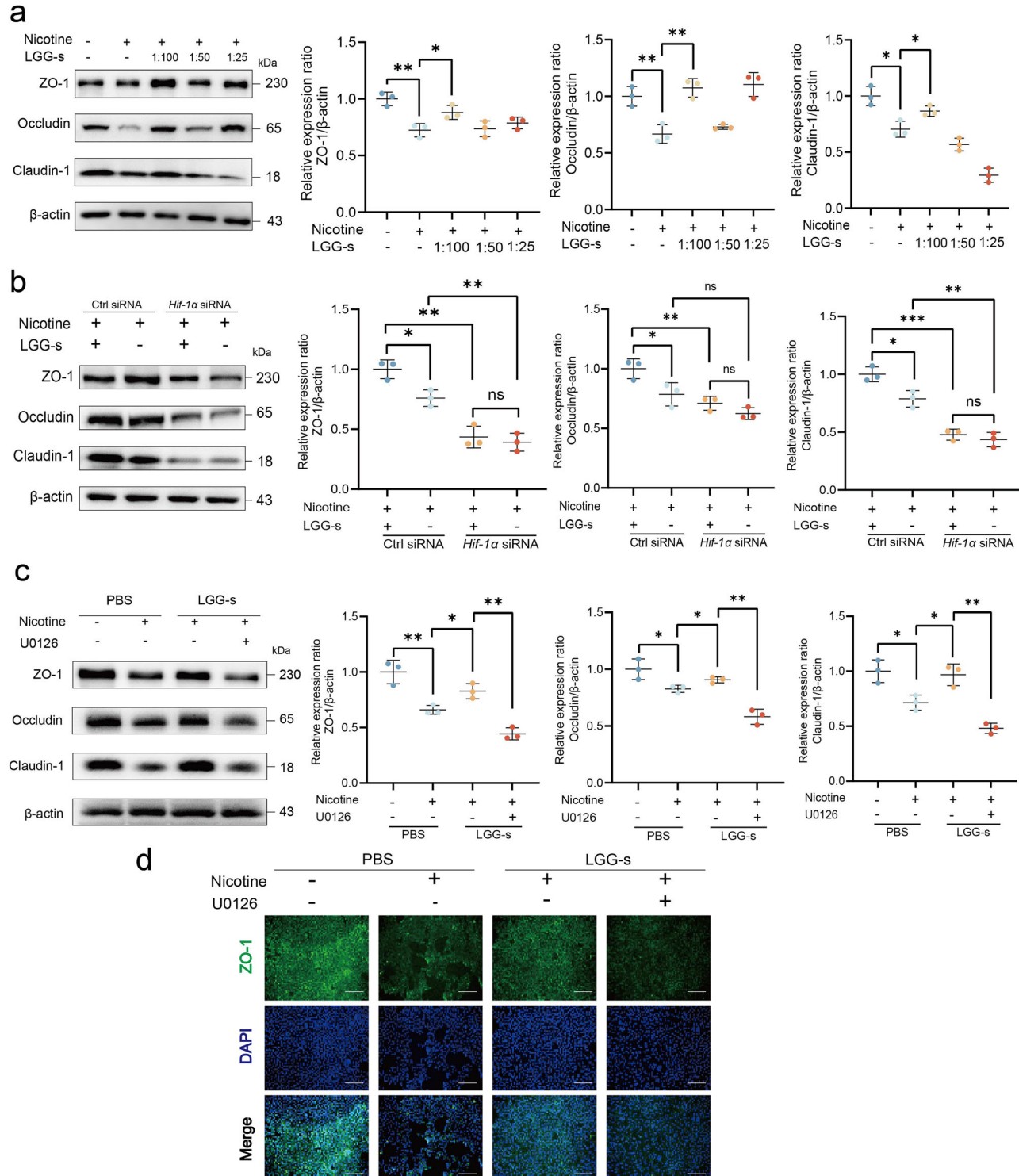

**Fig. 8 | HIF-1α/MEK/ERK pathways in LGG-s intestinal protection. a** Exploring the optimal concentration of LGG-s. **b** The role of LGG-s in *HIF-1α*-specific siRNA knockdown Caco-2 cell. **c** Western blot analysis for ZO-1, occludin, and claudin-1 normalized by β-actin after treatment with MEK inhibitor U0126 ($n = 3$). **d** IF staining of ZO-1. Scale bar = 200 μm. $^{*}P < 0.05$, $^{**}P < 0.01$, $^{***}P < 0.001$. Data are presented as mean ± SD.

the endogenous increase in intestinal *Akkermansia* levels and intestinal integrity warrants further exploration.

Intestinal barrier function was further evaluated in mice, revealing that nicotine exacerbates intestinal barrier damage in MASH. The expression of key tight junction proteins, including ZO-1, occludin, and claudin-1, exhibited a marked reduction in nicotine-exposed MASH mice. In addition to impairing the mechanical barrier of the intestine, nicotine also compromised the chemical and immune defenses. This was demonstrated by a significant reduction in the expression of antimicrobial peptides, including Defa2, Reg3b, Reg3d, Reg3g, Ang4, Lyz1, and Lyz2, as well as an upregulation of pro-inflammatory cytokines such as TNF-α, IL6, and IL1B. The underlying mechanism involved chronic nicotine exposure leading to intestinal microbiota dysbiosis, characterized by a decrease in beneficial bacteria and an increase in pathogenic microbes. This dysbiosis not only

disrupted intestinal immune homeostasis and enhanced intestinal permeability, resulting in a "leaky gut" phenotype[21], but also allowed harmful gut-derived metabolites, such as LPS, to translocate into the bloodstream. These changes drove systemic inflammation and accelerated the progression of MASH.

Abx-induced intestinal microbiota depletion, also referred to as germ-free or pseudo-germ-free mice, has been widely used to explore the role of the intestinal microbiota in host physiology[22]. Our results, based on α-diversity analysis, demonstrated that Abx treatment significantly reduced the diversity and richness of the intestinal microbiota, confirming that Abx treatment effectively depleted intestinal microbial populations. Our current study further indicated that nicotine exposure exacerbates liver and intestinal dysfunction in MASH mice, an effect that can be alleviated through microbiota depletion. These findings suggested that microbiota depletion was associated with a reduction in nicotine-induced damage. It was also possible that the depletion of the intestinal microbiota led to a reduction in certain harmful metabolites produced by nicotine, although this hypothesis required further investigation.

The intestinal epithelial cells were under a "physiological hypoxia" state at baseline, a condition that contributes to the stabilization of HIF-1α[23]. HIF-1α was the primary transcription factor that regulated the expression of multiple barrier-related genes in the intestine[24]. However, smoking disrupted this physiological hypoxic state, and the interaction between nicotine and HIF-1α may contribute to this disruption, further affecting intestinal function. Therefore, to investigate the role of HIF-1α in nicotine-induced MASH, we generated intestinal epithelial-specific *HIF-1α* knockdown mice. Following the knockdown of HIF-1α, the intestinal barrier function in nicotine-exposed MASH mice was further compromised. These findings suggested that the upregulation of HIF-1α may play an adaptive protective role in the intestine of nicotine-exposed MASH mice.

Probiotics can promote the growth and differentiation of intestinal epithelial cells, increase the production of short-chain fatty acids (SCFAs), and enhance tight junctions between cells, thereby strengthening the integrity of the mechanical barrier[25]. The LGG-s enriched with SCFAs regulates the expression of genes related to intestinal barrier function and promotes the repair of the intestinal barrier[13]. Additionally, previous study had shown that LGG-s can directly activate intestinal HIF-1α, and the upregulation of HIF-1α enhanced the expression of local antimicrobial peptides, promoting microbial homeostasis[26]. Consistent with our findings, probiotic supplementation alleviated intestinal damage in nicotine-exposed MASH; however, in intestinal *HIF-1α*-deficient mice, the protective effect of LGG supernatant enriched with SCFAs on nicotine-induced intestinal barrier damage in MASH was abolished. The activation of the MEK/ERK signaling pathway was essential for maintaining intestinal barrier integrity[27]. Our study revealed that in HIF-1α knockdown mice, nicotine exposure significantly reduced the phosphorylation levels of MEK/ERK in the intestines of MASH mice. Furthermore, when MEK was inhibited, LGG-s failed to restore intestinal barrier integrity.

In this study, we observed that LGG-s and HIF-1α activation alleviated liver injury and intestinal barrier dysfunction in nicotine-exposed MASH mice. However, these protective effects do not target the pathogenic effects of nicotine per se, but rather counteract the gut-liver axis disruption induced by nicotine. Previous studies from our team and others have demonstrated that LGG-s protects against alcohol-induced liver injury by restoring intestinal homeostasis and activating HIF-1α, rather than targeting alcohol directly[9,10]. Similar protective roles have been reported in other disease models, including metabolic disorders and autoimmune hepatitis[13,28]. These findings suggest that LGG-s and HIF-1α act through broadly conserved pathways that support gut-liver axis integrity. In our current model, nicotine exacerbated MASH pathology in the context of CDAHFD, and this was accompanied by intestinal barrier impairment and microbiota dysbiosis. Future studies are warranted to elucidate how nicotine directly promotes MASH progression, thereby informing the development of targeted therapeutic strategies.

One limitation of our study is the use of intraperitoneal injection as the mode of nicotine administration, which does not fully replicate nicotine exposure observed in human smokers. Unlike the continuous, low-dose absorption via inhalation, intraperitoneal injection produces non-physiological peaks in plasma nicotine levels. Previous studies have attempted oral administration or the use of Alzet minipumps, but these approaches were constrained by significant stress responses, as evidenced by elevated corticosterone levels—even in vehicle-treated animals[29]. Similarly, jugular vein cannulation caused stress regardless of nicotine exposure[29]. Given these challenges, we adopted the widely used intraperitoneal injection method. The selected dosing regimen (1.5 mg/kg, once daily) approximates the daily nicotine exposure of a heavy smoker[30]. While this method ensures experimental consistency and minimizes procedural stress, we acknowledge that hepatic first-pass metabolism may influence nicotine distribution and potentially confound the differentiation between intestinal and hepatic effects. Therefore, future studies using alternative delivery approaches such as administration via drinking water or smoke exposure may better simulate human nicotine intake patterns and further validate the gut-liver mechanistic axis.

Another limitation of our study is the use of a pseudo-germ-free mouse model generated through long-term administration of a broad-spectrum antibiotic cocktail. Although this is a widely accepted and practical method for depleting the gut microbiota[31,32], it may also induce systemic metabolic effects, such as alterations in bile acid metabolism and immune function, which could potentially influence the progression of MASH. Antibiotic treatment may introduce confounding factors. Future studies using germ-free mice will be necessary to more accurately determine the causal role of the microbiota in mediating the effects of nicotine.

In conclusion, our findings elucidated that nicotine exacerbated MASH by inducing intestinal microbiota dysbiosis and intestinal barrier disruption. HIF-1α regulated the intestinal barrier via the MEK/ERK pathway, and its deficiency exacerbated intestinal barrier dysfunction, thereby worsening nicotine-exposed MASH and leading to more severe intestinal dysbiosis and increased bacterial translocation. LGG-s supplementation alleviates liver damage in nicotine-exposed MASH, but is ineffective in the absence of HIF-1α. Our findings provide new insights into smoking-related MASH pathogenesis and highlight potential therapeutic targets.

## Methods
### Animal models
Male 8-week-old C57BL/6J mice weighing 20–22 g, obtained from Gaofei Biotechnology Co., Ltd (Wenzhou, China), were used for all experiments. Mice were randomly divided into four groups ($n = 6$ per group) in a pathogen-free facility under a controlled temperature (22 °C) and photoperiod (12-h light, 12-h dark cycle), with ad libitum access to water and food. Mice were fed either a purified control diet (1010086; 21.5% of calories from protein, 67.4% from carbohydrates, and 11.1% from fat; Jiangsu Xietong Pharmaceutical Bio-engineering, China) or a choline-deficient, L-amino acid-defined, high-fat diet (CDAHFD; A06071302; 60% of calories from fat, 0.1% methionine, and no added choline; Research Diets, Inc., New Brunswick, NJ, USA) for 10 weeks. Mice received intraperitoneal injections of nicotine (N3876, Sigma-Aldrich) 1.5 mg/kg body weight or saline once daily for 10 weeks[29,33]. Mice were fasted overnight before euthanasia at the end of the 10th week. This project was conducted at the Laboratory Animal Center of The First Affiliated Hospital of Wenzhou Medical University, with supervision and approval from the Institutional Animal Care and Use Committee of The First Affiliated Hospital of Wenzhou Medical University (WYYY-AEC-2023-154).

### Fecal microbiota analysis
The OMEGA Soil DNA Kit (D5625-01) (Omega Bio-Tek, Norcross, GA, USA) was used to extract genomic DNA from fecal samples, and the

purity and concentration of the DNA were assessed. The V3-V4 variable region of the 16S rRNA gene was amplified via PCR using region-specific primers 338F 5′-barcode+ACTCCTACGGGAGGCAGCA-3′ and 806R 5′-GGACTACHVGGGTWTCTAAT-3′, and a high-fidelity DNA polymerase. PCR products were analyzed using a 2% agarose gel, and target bands were excised and purified using the Quant-iT PicoGreen dsDNA Assay Kit. Sequencing was conducted on an Illumina NovaSeq machine using the NovaSeq 6000 SP Reagent Kit (500 cycles). Representative sequences of each Amplicon Sequence Variant (ASV) were selected using the QIIME 2 toolkit and aligned with the SILVA database (version 138) for taxonomic annotation. Classification was performed using the classify-sklearn algorithm with default parameters. Alpha and beta diversity indices were calculated using QIIME2 software. The Kruskal–Wallis rank-sum test was used to assess variations and differences among groups.

### RNA sequencing analysis

Total RNA was extracted from liver and ileum tissues using TRIzol reagent (TakaRa, Japan). The RNA samples were assessed for purity by measuring the A260/A280 absorbance ratio using a NanoDrop ND-2000 spectrophotometer (Thermo Scientific, USA) and for integrity by determining the RNA Integrity Number with an Agilent Bioanalyzer 4150 (Agilent Technologies, CA, USA). The PE library was prepared following the protocol of the ABclonal mRNA-seq Lib Prep Kit (ABclonal, China). To purify mRNA, 1 μg of total RNA was processed using oligo magnetic beads and subsequently fragmented in ABclonal First Strand Synthesis Reaction Buffer. Using the fragmented mRNA as a template, the first-strand cDNA was synthesized with random primers and reverse transcriptase, followed by the synthesis of the second-strand cDNA using DNA polymerase I, RNase H, buffer, and dNTPs. The resulting double-stranded cDNA fragments were ligated with adapter sequences for PCR amplification. The amplified products were purified and evaluated for quality using an Agilent Bioanalyzer 4150. Sequencing was performed on the NovaSeq 6000 platform with 150 bp paired-end reads. The raw sequencing data generated from the Illumina platform were used for bioinformatic analyzes. Differential expression analysis between groups was conducted using DESeq2, with the default thresholds for identifying differentially expressed genes (DEGs) set to $|log2FC| > 1$ and $Padj < 0.05$. Gene Ontology (GO) functional enrichment and Kyoto Encyclopedia of Genes and Genomes pathway enrichment analyzes were performed using the clusterProfiler R package, with significant enrichment determined at $P < 0.05$.

### Transmission electron microscopy

Freshly harvested mouse ileum tissues were immediately cut into 1 mm³ pieces and immersed in an adequate volume of fixation solution (2.5% glutaraldehyde). The tissues were fixed at room temperature in the dark for 2 h and then transferred to 4 °C for storage. The samples were further prepared, sectioned, and imaged using transmission electron microscopy (TEM). Electron micrographs were acquired using a Hitachi HT7800 TEM.

### Antibiotic cocktail treatment of mice

The antibiotic cocktail (Abx) consisted of ampicillin (1 g/L), neomycin (1 g/L), metronidazole (1 g/L), and vancomycin (0.5 g/L)[31]. Mice were randomly divided into four groups (n = 6 per group), labeled as CDAHFD, CDAHFD + Abx, CDAHFD + Nicotine, and CDAHFD + Nicotine + Abx. The CDAHFD and CDAHFD+Nicotine groups were provided with regular drinking water without antibiotics. Water bottles were replaced every 2 days. At the end of the 10th week, mice were fasted and euthanized. Blood, feces, liver, and ileum tissues were collected. The tissues were rapidly frozen in liquid nitrogen and stored at −80 °C for further analysis.

### AAV7-HIF-1α treatment of Mice

To investigate whether nicotine-induced intestinal dysfunction in MASH was associated with *HIF-1α* expression, *HIF-1α* gene expression was silenced using AAV7-*HIF-1α*. The experimental groups were as follows: (i) CDAHFD + AAV7-Vector, (ii) CDAHFD + AAV7-*HIF-1α*, (iii) CDAHFD + Nicotine + AAV7-Vector, and (iv) CDAHFD + Nicotine + AAV7-*HIF-1α*. The timing of AAV administration via tail vein injection was shown in Fig. 6e. At the end of week 13, mice from all four groups were euthanized, and blood and tissue samples were collected for further experimental analyzes.

### Culture of LGG and preparation of LGG-s

*Lactobacillus rhamnosus* GG (BNCC 134266; Henan, China) was cultured in MRS medium under microaerobic conditions at 37 °C for 24–48 h. The bacterial concentration reached $10^9$ colony-forming units per milliliter (CFU/mL), which was used for preparing Lactobacillus culture supernatant and strain cryopreservation. The culture supernatant (LGG-s) was obtained by filtering the bacterial culture through 0.22 μm filter. Mice in the treatment group were administered LGG-s at a dose of 1 mL/day via oral gavage, while control mice received an equal volume of normal saline.

### Serum analysis

Blood samples were centrifuged at 3000 rpm for 15 min to extract serum. Serum alanine aminotransferase (ALT) and aspartate aminotransferase (AST) levels were measured using an automated biochemical analyzer (Chemray 800, Rayto, Shenzhen, China). The remaining serum was used to determine LPS concentrations according to the instructions of the LPS assay kit (Boyun Shanghai China).

### Western blotting

Liver tissues, ileum tissues, and cell extracts were homogenized in ice-cold RIPA (P0013B, Beyotime, Shanghai, China) buffer containing protease and phosphatase inhibitors (P1260-1, Applygen, Beijing, China). Extract nuclear protein using nuclear protein extraction kit (EX2550, Solarbio, Beijing, China). An appropriate amount of protein was separated by SDS-PAGE and transferred onto polyvinylidene difluoride (PVDF) membranes (Millipore). After blocking with 5% nonfat milk, the PVDF membranes were incubated overnight at 4 °C with primary antibodies, including Zonula occludens-1 (ZO-1) (21773-1-AP, Proteintech, Wuhan, China), Occludin (91131S, CST), Claudin-1 (28674-1-AP, Proteintech), MEK1/2 (A4868, Abclonal, Wuhan, China), p-MEK1/2 (AP1349, Abclonal), ERK1/2 (11257-1-AP, Proteintech), p-ERK1/2 (ET1603-22, Huaan, Hangzhou, China), HIF-1α (36169T, CST), and β-actin (AF7018, Affinity, Jiangsu, China). The membranes were washed with TBST at room temperature, followed by incubation with secondary antibodies (LF102 and LF101, Epizyme, Inc., Shanghai, China) for 2 h at room temperature with gentle shaking. The PVDF membranes were then washed three times with TBST for 10 min each at room temperature. Protein bands were visualized using the Bio-Rad immunoblot detection system (Bio-Rad Laboratories, Hercules, CA, USA). Quantitative analysis of Western blot bands was performed using ImageJ software (Bethesda, MD, USA).

### Quantitative real-time PCR

Total RNA was extracted from liver and ileum tissues using the TRIzol reagent. Reverse transcription was performed to synthesize cDNA using the PrimeScript™ RT Reagent Kit with gDNA Eraser (RR047A, Takara, Beijing, China) following the protocol provided by the manufacturer. Quantitative polymerase chain reaction (qPCR) was conducted using a qPCR kit (Q712-02, Vazyme, Nanjing, China) on a real-time fluorescence PCR system (ABI 7500, Applied Biosystems, Waltham, MA, USA). The relative expression levels of target genes, normalized to *Actb*, were calculated using the $2^{-\Delta\Delta Ct}$ method. The primer sequences used in this study were provided in Supplementary Table 1.

### Histological analysis

Liver and ileum tissues from mice were either fixed in 4% paraformaldehyde or embedded in OCT compound. Paraffin-embedded sections of liver and

ileum tissues were stained with hematoxylin and eosin (H&E) to assess lipid accumulation and with Masson's trichrome to evaluate hepatic fibrosis. Frozen liver sections were stained with Oil Red O to visualize lipid droplets. All procedures were carried out according to standard protocols, followed by microscopic examination. AB-PAS staining was performed on paraffin-embedded ileal sections to visualize goblet cells.

## Immunofluorescence
Liver and ileum tissues or cells were fixed with 4% paraformaldehyde. After blocking with 5% bovine serum albumin (BSA), the samples were incubated overnight at 4 °C with primary antibodies against CD20 (GB11540, Servicebio, Wuhan, China), F4/80 (A1256, Abclonal), Claudin-1 (28674-1-AP, Proteintech), and ZO-1 (21773-1-AP, Proteintech, Wuhan, China). The sections were then incubated with secondary antibodies at 25 °C in the dark for ~1 h, followed by counterstaining with DAPI. Finally, the samples were examined using a laser scanning confocal microscope (BZ-X800, Keyence, Japan).

## Immunohistochemistry
Paraffin-embedded ileum tissue sections were deparaffinized, subjected to antigen retrieval, and blocked, followed by overnight incubation at 4 °C with an anti-Ly6G antibody (GB11229, Servicebio, Wuhan, China). After incubation with a secondary antibody, enzyme conjugation, and DAB staining, the sections were counterstained with hematoxylin. Images were acquired using a light microscope (Olympus, Japan).

## Cell cultures and transient transfection
Human colorectal cancer Caco-2 cells (iCell-h032, iCell Bioscience, Shanghai, China) were cultured in MEM medium (PM150410, Procell, Wuhan, China) supplemented with 10% fetal bovine serum and 1% penicillin-streptomycin solution (Solarbio, Beijing, China) at 37 °C in a humidified incubator with 5% $CO_2$. To simulate intestinal conditions in MASLD, cells were treated with 1 mM free fatty acid in 1% BSA for 24 h[34]. Cells were pretreated with LGG-s at three gradient concentrations (1:100, 1:50, and 1:25) for 3 h, followed by treatment with nicotine and U0126 for 24 h[13]. Based on the CCK8 assay, the concentrations of nicotine and U0126 were maintained at 1 mM and 20 μM, respectively. HIF-1α-specific siRNA and non-targeting control siRNA were transfected into cells using Lipofectamine 2000 reagent (Thermo Fisher Scientific) according to the manufacturer's protocol. HIF-1α-specific siRNA (5′-GCUGGAGACA-CAAUCAUAUTT-3′, 5′-AUAUGAUUGUGUCUCCAGCGG-3′) and non-targeting control siRNA (5′-UUCUCCGAACGUGUCACGUdTdT-3′, 5′-ACGUGACACGUUCGGAGAAdTdT-3′) were synthesized by Guannan Biotechnology Co., Ltd (Hangzhou, China).

## Statistics and reproducibility
All data were analyzed using GraphPad Prism 9.0 and presented as mean ± standard deviation. Comparisons between two groups were performed using either the unpaired Student's t-test or the Mann–Whitney U test. For comparisons involving multiple groups, one-way or two-way analysis of variance (ANOVA) followed by Tukey's post hoc test was employed. P-value < 0.05 was considered statistically significant. Sample sizes and number of replicates are reported in the Figure Legends.

## Ethics
This Project has been completed in Laboratory Animal Center of The First Affiliated Hospital of Wenzhou Medical University, and has been supervised and approved by Institutional Animal Care and Use Committee of The First Affiliated Hospital of Wenzhou Medical University (WYYY-AEC-2023-154).

## Data availability
Raw data have been deposited to National Center for Biotechnology Information under the BioProject number PRJNA1284962. (https://dataview.ncbi.nlm.nih.gov/object/PRJNA1284962?reviewer= 9en889vvqajdhmbuhplljrs038). The source data for graphs are provided in Supplementary Data 1. Uncropped/unedited blots are provided in the Supplementary information as Supplementary Fig. 6. All data generated or analyzed during this study are available from the lead contact on reasonable request. All data are available from the corresponding authors upon reasonable request.

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

## Acknowledgements

This research was funded by the National Natural Science Foundation of China (82070593) and Zhejiang Provincial Natural Science Foundation of China under Grant (LTGY24H030010). This research was supported by Key R&D Program of Zhejiang (2022C03125) and the 2025 Zhejiang Provincial Major Research and Development Project (Leading Goose): New Technology Research on Microencapsulated Fecal Microbiota Transplantation for the Treatment of Autoimmune Liver Disease Based on Targeted Delivery. Project Number: 2025C02135. We are grateful for the Shanghai Applied Protein Technology Co., Ltd Shanghai, China for 16S rRNA and RNA sequencing.

## Author contributions

Conceptualization: F.F. Yi, D.Z. Chen, and Y.P. Chen; methodology: F.F. Yi, J.Y. Wang, Y. Chen, and Y.H. Xia; investigation: F.F. Yi, H.F. Zhou, and Y.J. Shen.; formal analysis: F.F. Yi, J.Y. Wang, and H.F. Zhou; validation: F.F. Yi, J.Y. Wang, Y. Chen, and Y.H. Xia; resources: F.F. Yi, Y.L. Huang, S.Z. Fang, X.D. Wang, and Y. Zhang; writing—original draft preparation: F.F. Yi; supervision: X.D. Wang, Y. Zhang, D.Z. Chen, and Y.P. Chen; funding acquisition: X.D. Wang, D.Z. Chen, and Y.P. Chen. All authors have read and agreed to the published version of the manuscript.

## Competing interests

The authors declare no competing interests.
