## [Transparent Peer Review file · Communications Biology]

Nicotine exacerbates MASH via inducing intestinal dysbiosis and barrier dysfunction

Corresponding Author: Professor Yongping Chen

Version 0:

Reviewer comments:

Reviewer #1

(Remarks to the Author)

The authors have investigated the effects of nicotine on liver inflammation, microbial dysbiosis, and intestinal barrier dysfunction in diet-induced MASH model mice. They have clarified that administration of nicotine aggravates hepatic steatosis and inflammation in MASH mice. Nicotine affects gut microbial composition and the expression of tight junction-associated molecules. Additionally, nicotine exposure induces HIF-1alpha expression via MEK/ERK pathway, which is associated with the inhibitory effect of probiotic *Lactobacillus* on MASH. These findings are interesting, and there has been an increasing interest in the gut-liver axis as a promising therapeutic target for MASH. However, several problems are to be clarified in the present study, and some data are required to support their theory.

Major

1. The detailed mechanism by which nicotine aggravates hepatic steatosis has not been fully elucidated. The authors showed that nicotine increased hepatic TG and cholesterol levels even in control mice. In addition, fat accumulation was greater in the CDHFD+NIC+Abx than that in CDHFD+Abx as shown in Fig 5. Do these results suggest that the effect of nicotine in inducing hepatic steatosis occurs independently of changes in the intestinal flora?
2. As shown in Fig. 3L, M, N, the cytokine levels in the intestine of the CDHFD+NIC mice were higher than those of the CDHFD mice. Does the administration of antibiotics attenuate the increased cytokine expression? Is it possible that nicotine directly affects the function of immune cells in the intestinal tract? If so, wouldn't it be necessary to verify the effect of nicotine on immune cells in addition to analyzing epithelial cells?
3. The mucus secreted by goblet cells is a crucial factor in the mucosal barrier function. Many reports have shown that *Akkermansia* increases mucus production in the intestinal mucosa. The authors should elucidate the effect of nicotine on mucin production by goblet cells.
4. The mechanism of HIF-1alpha induction in MASH has been still elucidated. If the increased expression of HIF-1alpha is due to an anaerobic environment caused by changes in the gut flora in CDHFD+NIC, would the expression levels of HIF-1alpha be counteracted by administering antibiotics? Additionally, wouldn't it be appropriate to extract the nuclear fraction and confirm the amount of HIF-1alpha?

Minor

1. Please provide more details about the "intestinal tissues" used for PCR and western blotting. Do intestinal tissues refer to the entire small intestine or do they include the colon? Were Peyer's patches removed?
2. Please revise Fig. 6G and 6K.

Reviewer #2

(Remarks to the Author)

The manuscript by Yi et al. demonstrates that nicotine exposure can exacerbate choline-deficient, high-fat diet (CDHFD)-induced MASH in mice by (i) promoting gut microbiota dysbiosis and (ii) increasing gut barrier permeability. The authors

then showed that CDHFD-induced MASH could be corrected by intervening with antibiotics, and also by gavaging the mice with the culture supernatant of *Lactobacillus rhamnosus* (LGG-s). The protection conferred by LGG-s was lost when intestinal hypoxia-inducible factor 1 alpha (Hif-1a) was knocked down. The manuscript is straightforward, but there are some concerns that need to be addressed.

Major comment:

1. The manuscript is heavily inspired from the corresponding author's prior work on intestinal Hif-1a and *L. rhamnosus*, published in *J Hepatol* 2018 (PMID: 29803899). Yet, the current manuscript did not give proper credit/discussion for that prior work. Because of that, it is difficult to understand why the authors target Hif-1a and *L. rhamnosus* (when the authors clearly had the prior experience). The paper on "*Lactobacillus rhamnosus* GG treatment potentiates intestinal hypoxia-inducible factor" (PMID: 22093263) should be cited as well.
2. Why the authors decided to oral gavage the culture supernatant of *L. rhamnosus*, but not gavage the bacteria itself? It is also not clear what is in the culture supernatant that mediate the protection.
3. Nicotine metabolism occurs predominantly in the liver. Thus, a major weakness is that the manuscript did not rule out the direct effect of nicotine (i.e., administered intraperitoneally) on the liver. The authors should at least consider it in the discussion.
4. I find the writing on the 'nicotine - Hif-1a - *L. rhamnosus*' link to be misleading (albeit unintentionally). From the data, I am convinced that nicotine can exacerbate CDHFD-induced MASH. However, it is not clear whether nicotine has any interaction with Hif-1a and *L. rhamnosus*. Even without nicotine treatment, (i) antibiotics treatment protect against MASH, (ii) Hif-1a knockdown aggravate MASH, and (iii) *L. rhamnosus* culture supernatant alleviate MASH. This led me to reason that antibiotics, Hif-1a and *L. rhamnosus* are interacting directly with the MASH model, and not by mitigating the effects of nicotine.

In addition, the corresponding author had previously shown that Hif-1a and *L. rhamnosus* protected against alcohol-induced liver injury via their effects on gut barrier function. This again led me to believe that the protection seen with Hif-1a and *L. rhamnosus* is 'generalizable' and not specific to the effects of nicotine. These points should be addressed or discussed.

5. AAV7-Hif-1a treatment may have off-target effects, and therefore should be validated with the intestinal-specific Hif-1a knockout mouse model. This mouse model is mentioned in the Discussion (and also employed previously in *J Hepatol* 2018). However, the data is missing in the manuscript.

Minor comment:

6. In Fig. 6G, it is not clear which group received the AAV7 vector vs AAV7-Hif-1a.
7. This sentence on "Nicotine exacerbates MASH severity via HIF-1 α knockdown in mice" is misleading. The data does not show that nicotine can knockdown HIF-1a.
8. The abstract mentioned "intestinal permeability facilitated bacterial translocation to the liver". However, no such data was shown in the manuscript.
9. It is not clear whether the control diet 1010086 is a purified diet, and whether it is isocaloric to CDHFD. This need to be stated clearly. One criticism with the use of rodent chow (non-purified diet) is because it contains other plant-derived ingredients and fibers (i.e., not matched to CDHFD) and therefore could confound the microbiota analyses. This is a minor comment because the main analyses compare CDHFD to CDHFD+nicotine, which is appropriate.

Reviewer #3

(Remarks to the Author)

1. The major claims of the paper is that this study highlights the pivotal role of nicotine in exacerbating MASH through intestinal microbiota modulation and barrier dysfunction mediated via HIF-1 α regulation.
2. This study is quite intriguing, as it scientifically explains the harmful effects of smoking on the human body, captures public attention, and guides a broad population towards adopting healthier habits. The findings are novel in linking nicotine-induced intestinal dysbiosis and HIF-1 α -mediated barrier dysfunction to MASH progression.
3. In the part of animal models, please develop a flowchart illustrating experimental animal grouping and feeding protocols to improve clarity and visual representation.
4. The manuscript is methodologically robust, employing a combination of in vivo models (CDHFD diet + nicotine exposure), multi-omics approaches (16S rRNA sequencing, RNA-seq), and mechanistic studies (AAV-mediated HIF-1 α knockdown, MEK/ERK pathway inhibition). However, several technical limitations should be addressed: (1) While antibiotic (Abx) treatment effectively depletes microbiota, prolonged use may induce metabolic side effects (e.g., altered bile acid metabolism, immune modulation) that could confound results. The study does not account for these systemic effects, which may indirectly influence MASH progression. (2) The AAV7-HIF-1 α knockdown model lacks validation of tissue specificity

(e.g., intestinal vs. hepatic HIF-1 α suppression). Off-target effects or incomplete knockdown could affect interpretation.
5. This work is highly relevant to hepatology, gastroenterology, and microbiome research. Key strengths include:
Translational Implications: The gut-liver axis is a therapeutic hotspot for metabolic diseases. Demonstrating nicotine's role in disrupting this axis via HIF-1 α offers actionable targets (e.g., probiotics, HIF-1 α modulators).
Public Health Impact: Smoking is a modifiable risk factor for MASLD/MASH. This study provides mechanistic evidence to support smoking cessation campaigns and probiotic interventions for smokers with liver disease.
6. To enhance rigor and impact for the paper, if possible, please supplement HIF-1 α Rescue Experiments: Treat HIF-1 α -deficient mice with HIF-1 α agonists (e.g., FG-4592) to confirm reversibility of nicotine-induced damage.
7. In terms of language, the introduction needs more in-depth description in order to attract readers' attention.
In summary, the manuscript presents compelling evidence that nicotine exacerbates MASH via gut microbiota dysbiosis and HIF-1 α -dependent barrier dysfunction.

Version 1:

Reviewer comments:

Reviewer #1

(Remarks to the Author)

The authors have thoroughly addressed the reviewers' comments and made significant improvements to the manuscript. I recommend that the manuscript be accepted for publication.

Reviewer #2

(Remarks to the Author)

The authors have addressed my comments, and I have only minor comments to point out.

1. The term "pathogenic" should be changed to "pathologic" in the Abstract section. This is because nicotine is not a pathogen.

2. In the Abstract, the sentence on "...barrier dysfunction mediated by HIF-1 α " is misleading, considering the author's premise that HIF-1 α plays a protective role.

3. Please provide method and details on goblet cell staining.

4. RNA-seq detect changes in epithelial cells, but not the gut microbiota. The following sentence in Line 388 was incorrect. Line 388: "RNA sequencing analysis was performed on intestinal tissue samples to determine potential alterations in the intestinal microbiota of CDAHFD mice induced by nicotine exposure."

I have also reviewed Reviewer 3's comments and the authors responses.

Herein I summarize Reviewer 3's comments, and provide my opinion on whether the authors have adequately addressed those comments.

Point #1: Reviewer 3 asked for a flow chart for animal studies.

#The authors had provided the flowchart in Fig. 1.

Point #2: Reviewer 3 asked the authors to discuss the limitation of antibiotics use.

#The authors addressed this point by adding a new paragraph in the Discussion section.

Point #3: Reviewer 3 commented on the use of AAV7-HIF-1 α , and asked the authors to discuss the possibility of incomplete knockdown.

#The authors provided additional reviewer-only data, confirming that the degree of knockdown is satisfactory. The data in the manuscript also provided sufficient support that the degree of knockdown was enough to yield a change in the MASH phenotype.

Point #4: Reviewer 3 asked for the validation of tissue specificity (e.g., intestinal vs. hepatic HIF 1 α suppression).

#The authors' response was not satisfactory. I agree with Reviewer 3 and will ask the authors to provide western blot and qRT-PCR data for HIF-1 α in the liver (authors only showed data for intestine).

Point #5: Reviewer 3 suggested the authors to treat HIF 1 α deficient mice with HIF 1 α agonists

#The authors reasoned that such approach is beyond the scope of the study and capability of the authors at this time. I agree with the authors.

Point #6: Reviewer 3 suggested the authors to revise the writing on Introduction.

#The authors had addressed this point.

Summary:

I find the authors' response to Reviewer 3's comments to be overall satisfactory, except on the need to validate the tissue specificity of AAV7-mediated HIF-1 α knockdown. Whether or not there is an unintended non-specific knockdown of HIF-1 α in the liver, the study conclusion would remain intact. Nonetheless, such data is important to maintain scientific rigor. The authors need to provide western blot and qRT-PCR data of liver HIF-1 α (in the same figure where they had shown HIF-1 α expression in the intestine) after AAV7-HIF-1 α treatment.

Reviewer #1

Comment 1: *The detailed mechanism by which nicotine aggravates hepatic steatosis has not been fully elucidated. The authors showed that nicotine increased hepatic TG and cholesterol levels even in control mice. In addition, fat accumulation was greater in the CDAHFD+NIC+Abx than that in CDAHFD+Abx as shown in Fig 5. Do these results suggest that the effect of nicotine in inducing hepatic steatosis occurs independently of changes in the intestinal flora?*

Response 1:

Thank you for your valuable comments. Under control diet (Ctrl) conditions, nicotine administration led to an increase in TG and TC levels. However, histological analysis showed no apparent hepatic steatosis, and serum ALT and AST levels remained within the normal range. These findings suggest that in the context of metabolic homeostasis, nicotine alone induces mild disturbances in lipid metabolism but does not cause significant liver injury. In contrast, under CDAHFD conditions, nicotine markedly aggravated hepatic steatosis and significantly increased serum ALT and AST levels, indicating that its hepatotoxic effects are more pronounced in a metabolically stressed host.

Importantly, in the CDAHFD+NIC+Abx group, hepatic lipid accumulation was still observed compared to the CDAHFD+Abx group, although to a lesser extent than in the CDAHFD+NIC group. These results indicate that while nicotine has a direct effect on hepatic lipid metabolism, the gut microbiota plays a crucial amplifying role in mediating nicotine-induced hepatic steatosis.

Therefore, we believe that nicotine exacerbates MASH through multi-organ and cross-system mechanisms, which may include both direct effects on the liver and indirect effects mediated by the gut microbiota and intestinal barrier dysfunction. This study mainly focuses on the contribution of the gut-liver axis, and future investigations are needed to further clarify the relative importance of these distinct mechanisms.

Comment 2: *As shown in Fig. 3L, M, N, the cytokine levels in the intestine of the CDAHFD+NIC mice were higher than those of the CDAHFD mice. Does the administration of antibiotics*

attenuate the increased cytokine expression? Is it possible that nicotine directly affects the function of immune cells in the intestinal tract? If so, wouldn't it be necessary to verify the effect of nicotine on immune cells in addition to analyzing epithelial cells?

Response 2:

Thank you for your insightful comments. We have added qPCR analyses about the administration of antibiotics attenuate the increased cytokine expression in the revised manuscript in the “Results” section (Line 423-425). **Revised text reads:** In the nicotine-exposed CDAHFD group, intestinal pro-inflammatory factors were significantly reduced after Abx treatment (Figure 5g-i).

To further investigate the potential effect of nicotine on intestinal immune cells, we performed immunofluorescence staining for F4/80 and CD3 in the “Results” section (Line 377-379). **Revised text reads:** F4/80 and CD3 immunofluorescence staining showed increased infiltration of T lymphocytes and macrophages in nicotine-exposed CDAHFD mice (Supplementary Fig. 3).

Supplementary Fig. 3 F4/80 and CD3 staining of the ileal tissues. Scale bar = 200 μm . ***P < 0.001.

Comment 3: *The mucus secreted by goblet cells is a crucial factor in the mucosal barrier function. Many reports have shown that Akkermancia increases mucus production in the intestinal mucosa. The authors should elucidate the effect of nicotine on mucin production by goblet cells.*

Response 3:

Thank you for your valuable suggestion. We performed Alcian blue/periodic acid-Schiff (AB-PAS) staining on the ileum of mice to evaluate goblet cell numbers, as described in the Results section (Lines 370–373).

Revised text reads: The mucus secreted by goblet cells plays a critical role in maintaining mucosal barrier function. Nicotine exposure reduced the number of goblet cells in the intestinal mucosa, with a more pronounced reduction observed in the CDAHFD+NIC group (Supplementary Fig. 2).

Supplementary Fig. 2 Goblet cell staining and quantification of the ileal tissues. Arrows denote Goblet cells. Scale bar = 200 μm . *P < 0.05, **P < 0.01.

Comment 4: *The mechanism of HIF-1alpha induction in MASH has been still elucidated. If the increased expression of HIF-1alpha is due to an anaerobic environment caused by changes in the gut flora in CDAHFD+NIC, would the expression levels of HIF-1alpha be counteracted by administering antibiotics? Additionally, wouldn't it be appropriate to extract the nuclear fraction and confirm the amount of HIF-1alpha?*

Response 4:

Thank you for your insightful comments. We performed additional experiments to address these important points. Specifically, to explore whether the upregulation of HIF-1 α is associated with intestinal microbiota-induced alterations in the intestinal environment, we administered antibiotics (Abx) to deplete the intestinal microbiota in CDAHFD+NIC mice. We also extracted nuclear protein fractions using a nuclear protein extraction kit (EX2550, Solarbio, Beijing, China) and performed Western blot analysis to assess the transcriptionally active nuclear HIF-1 α .

Our results showed that nuclear HIF-1 α levels were significantly reduced after Abx treatment, supporting the hypothesis that intestinal microbiota alterations contribute to HIF-1 α induction in this context. These findings suggest that the transcriptional activation of HIF-1 α was attenuated under microbiota-depleted conditions, possibly due to alterations in the intestinal anaerobic environment.

Revised text reads:

In the section of Materials and Method (Lines 224-225): Extract nuclear protein using nuclear protein extraction kit (EX2550, Solarbio, Beijing, China).

In the section of Results (Lines 447-456): Study has demonstrated that HIF-1 α plays a key role in maintaining intestinal barrier integrity(17). Western blot analysis showed a marked upregulation of HIF-1 α expression in nicotine-exposed MASH mice (Fig. 6a-b). We speculate that nicotine exposure may exacerbate the intestinal anaerobic environment in MASH mice, which could contribute to the upregulation of HIF-1 α . To test this, we administered Abx to deplete the intestinal microbiota. Nuclear protein fractions were subsequently extracted and subjected to Western blot analysis. The results showed that nuclear HIF-1 α levels were significantly reduced after Abx treatment, supporting our hypothesis and indicating that the transcriptional activation of

HIF-1 α was attenuated under microbiota-depleted conditions (Fig. 6c-d).

Figure 6. HIF-1 α knockdown exacerbates nicotine-exposed MASH in mice. a, b, Western blot analysis for Hif-1 α normalized by β -actin. c, d, Western blot analysis for Hif-1 α normalized by β -actin after antibiotic cocktail (Abx).

Comment 5: Please provide more details about the “intestinal tissues” used for PCR and western blotting. Do intestinal tissues refer to the entire small intestine or do they include the colon? Were Peyer’s patches removed?

Response 5:

Thank you for your insightful comments. In our study, the “intestinal tissues” used for PCR and western blotting refer specifically to the ileum, based on previous research indicating that nicotine accumulates most abundantly in the ileum (*Nature*. 2022 Oct;610(7932):562-568. PMID: 36261549). These details have been revised to the Materials and Method section (Lines 222, 243) in the revised manuscript.

We did not remove Peyer’s patches during tissue collection. This decision was made to preserve the physiological integrity of the intestinal tissue, particularly in the context of studying host–microbiota interactions and immune-related signaling pathways (*Nat Rev Immunol*. 2014 Oct;14(10):667-85). As Peyer’s patches are an integral part of the gut-associated lymphoid tissue, they may contribute to the local immune microenvironment and mucosal barrier function, both of

which are relevant to our investigation of HIF-1 α expression and intestinal barrier integrity in MASH. Therefore, we chose to retain Peyer's patches to avoid potentially disrupting the native intestinal immune landscape. Importantly, Peyer's patches were consistently retained across all experimental groups, ensuring comparability of results.

Comment 6: Please revise Fig. 6G and 6K.

Response 6:

Thank you for pointing out the labeling error in Fig. 6G and 6K. We have carefully revised and corrected the figure legends. Due to supplementary experiments, the original **Figure 6G** and **6K** were changed to **Figure 6i** and **6m**. The updated figure has been included in the revised manuscript.

Reviewer #2

Comment 1: *The manuscript is heavily inspired from the corresponding author's prior work on intestinal Hif-1 α and *L. rhamnosus*, published in *J Hepatol* 2018 (PMID: 29803899). Yet, the current manuscript did not give proper credit/discussion for that prior work. Because of that, it is difficult to understand why the authors target Hif-1 α and *L. rhamnosus* (when the authors clearly had the prior experience). The paper on "Lactobacillus rhamnosus GG treatment potentiates intestinal hypoxia-inducible factor" (PMID: 22093263) should be cited as well.*

Response 1:

Thank you for your insightful and constructive comments. We sincerely apologize for not providing sufficient discussion of our previous work on intestinal Hif-1 α and *Lactobacillus rhamnosus* GG (*J Hepatol* 2018, PMID: 29803899). We acknowledge that this prior study provided important conceptual and mechanistic groundwork for the current manuscript. In the revised manuscript, we have added the relevant discussion to the Introduction section (Lines 103-107). **Revised text reads:** Our previous study demonstrated that mice with intestinal epithelial-specific knockout of HIF-1 α exhibited more severe liver injury and higher serum lipopolysaccharides (LPS) levels in a murine model of alcoholic liver disease, further confirming the critical role of HIF-1 α in maintaining intestinal microbiota homeostasis and gut barrier integrity(10).

We also sincerely appreciate your recommendation of the study (*Am J Pathol.* 2011 Dec;179(6):2866-75. PMID: 22093263). This is an excellent piece of work that aligns closely with the focus of our manuscript, and we have now included a discussion of this study in the Introduction section (Line 97-100). **Revised text reads:** Moreover, *Lactobacillus rhamnosus* GG enhances hypoxia-inducible factor 2 α signaling and upregulates the expression of intestinal tight junction proteins, thereby preserving intestinal barrier integrity and alleviating alcoholic liver injury(9).

Comment 2: *Why the authors decided to oral gavage the culture supernatant of *L. rhamnosus*, but not gavage the bacteria itself? It is also not clear what is in the culture supernatant that*

mediate the protection.

Response 2:

Thank you for your insightful comments. In our study, we chose to administer the culture supernatant of *Lactobacillus rhamnosus* (LGG-s) rather than the live bacteria itself because LGG-s contains a variety of bioactive metabolites secreted by LGG, including abundant short-chain fatty acids. Our previous research has demonstrated that LGG-s can effectively alleviate autoimmune liver injury (*Cell Death Dis.* 2023 Jan 28;14(1):68. PMID: 36709322).

In addition, immunosuppression is a major risk factor for *Lactobacillus*-induced bacteremia (*Cureus.* 2020 Feb 5;12(2):e6887. PMID: 32190450). Nicotine is a highly toxic compound, and prolonged exposure in mice may trigger systemic inflammatory responses (*Am J Physiol Regul Integr Comp Physiol.* 2018 Jun 1;314(6):R834-R847. PMID: 29384700). To minimize the potential risk, we opted for the safer approach of using LGG culture supernatant.

Nevertheless, we acknowledge the importance of this question and will consider exploring the therapeutic effects of live LGG administration in nicotine-exposed MASH mice in future studies.

Comment 3: *Nicotine metabolism occurs predominantly in the liver. Thus, a major weakness is that the manuscript did not rule out the direct effect of nicotine (i.e., administered intraperitoneally) on the liver. The authors should at least consider it in the discussion.*

Response 3:

Thank you for your important and constructive comments. We agree that the potential hepatic effects of intraperitoneally administered nicotine warrant consideration. In the revised manuscript, we have addressed this issue in the Discussion section as a limitation of our study (Lines: 632-648).

Revised text reads: One limitation of our study is the use of intraperitoneal injection as the mode of nicotine administration, which does not fully replicate nicotine exposure observed in human smokers. Unlike the continuous, low-dose absorption via inhalation, intraperitoneal injection

produces non-physiological peaks in plasma nicotine levels. Previous studies have attempted oral administration or the use of Alzet minipumps, but these approaches were constrained by significant stress responses, as evidenced by elevated corticosterone levels—even in vehicle-treated animals(13). Similarly, jugular vein cannulation caused stress regardless of nicotine exposure(13). Given these challenges, we adopted the widely used intraperitoneal injection method. The selected dosing regimen (1.5 mg/kg, once daily) approximates the daily nicotine exposure of a heavy smoker(33). While this method ensures experimental consistency and minimizes procedural stress, we acknowledge that hepatic first-pass metabolism may influence nicotine distribution and potentially confound the differentiation between intestinal and hepatic effects. Therefore, future studies using alternative delivery approaches such as administration via drinking water or smoke exposure may better simulate human nicotine intake patterns and further validate the gut-liver mechanistic axis.

Comment 4: *I find the writing on the “nicotine - Hif-1 α - L. rhamnosus” link to be misleading (albeit unintentionally). From the data, I am convinced that nicotine can exacerbate CDAHFD-induced MASH. However, it is not clear whether nicotine has any interaction with Hif-1 α and L. rhamnosus. Even without nicotine treatment, (i) antibiotics treatment protect against MASH, (ii) Hif-1 α knockdown aggravate MASH, and (iii) L. rhamnosus culture supernatant alleviate MASH. This led me to reason that antibiotics, Hif-1 α and L. rhamnosus are interacting directly with the MASH model, and not by mitigating the effects of nicotine.*

In addition, the corresponding author had previously shown that Hif-1 α and L. rhamnosus protected against alcohol-induced liver injury via their effects on gut barrier function. This again led me to believe that the protection seen with Hif-1 α and L. rhamnosus is ‘generalizable’ and not specific to the effects of nicotine. These points should be addressed or discussed.

Response 4:

Thank you for your thoughtful and insightful comments. We agree that the protective effects of *Lactobacillus rhamnosus* culture supernatant (LGG-s) and HIF-1 α may not be specific to nicotine exposure per se, but rather reflect more general mechanisms related to gut barrier maintenance and host-microbiota interactions. In our previous studies, we demonstrated that LGG-s ameliorates

alcohol-induced liver injury primarily by enhancing intestinal HIF signaling and restoring gut microbiota composition (*J Hepatol.* 2018, PMID: 29803899; *Am J Pathol.* 2011, PMID: 22093263). These effects were not directed at alcohol itself, but occurred through modulation of the intestinal environment, which appears to be a common mechanism across several liver injury models, including metabolic disease (*J Nutr Biochem.* 2020, PMID: 31760308) and autoimmune hepatitis (*Cell Death Dis.* 2023, PMID: 36709322).

In our current study, nicotine aggravated CDAHFD-induced MASH, and this was accompanied by gut microbiota dysbiosis and intestinal barrier disruption. Although the interventions with LGG-s and HIF-1 α modulation were applied in the context of nicotine exposure, we do not intend to imply a direct or specific interaction along a linear “nicotine–HIF-1 α –LGG” axis. Rather, we propose that the beneficial effects of LGG-s and HIF-1 α are due to their ability to restore gut-liver axis integrity, which is disrupted in various liver injury models, including but not limited to nicotine-induced MASH.

We have now clarified this point in the revised Discussion section (Lines: 617-630). In future studies, we aim to further investigate whether nicotine interacts directly with HIF-1 α or *Lactobacillus rhamnosus* at the molecular level, or whether these effects are primarily mediated through upstream changes in the gut microenvironment.

Revised text reads: In this study, we observed that LGG-s and HIF-1 α activation alleviated liver injury and intestinal barrier dysfunction in nicotine-exposed MASH mice. However, these protective effects do not target the pathogenic effects of nicotine per se, but rather counteract the gut-liver axis disruption induced by nicotine. Previous studies from our team and others have demonstrated that LGG-s protects against alcohol-induced liver injury by restoring intestinal homeostasis and activating HIF-1 α , rather than targeting alcohol directly(9,10). Similar protective roles have been reported in other disease models, including metabolic disorders and autoimmune hepatitis(16,32). These findings suggest that LGG-s and HIF-1 α act through broadly conserved pathways that support gut-liver axis integrity. In our current model, nicotine exacerbated MASH pathology in the context of CDAHFD, and this was accompanied by intestinal barrier impairment and microbiota dysbiosis. Future studies are warranted to elucidate how nicotine directly promotes MASH progression, thereby informing the development of targeted therapeutic strategies.

Comment 5: *AAV7-Hif-1 α treatment may have off-target effects, and therefore should be validated with the intestinal-specific Hif-1 α knockout mouse model. This mouse model is mentioned in the Discussion (and also employed previously in *J Hepatol* 2018). However, the data is missing in the manuscript.*

Response 5:

Thank you for highlighting the inconsistency between the Discussion section and the experimental approach used in our study regarding the HIF-1 α model. We sincerely apologize for the confusion caused by the unintended reference to a “knockout” model. In the original manuscript, we mistakenly referred to our AAV7-mediated knockdown strategy as a "knockout," which may have led to the misunderstanding that we employed an intestinal-specific Hif-1 α knockout mouse model. This terminology has now been corrected in the revised Discussion (Lines: 595, 613) to accurately reflect our use of AAV7-shHif-1 α -mediated gene silencing rather than genetic deletion.

We fully acknowledge that AAV7-mediated knockdown does not replicate the efficiency or intestinal specificity of a genetically engineered knockout model and may carry a risk of off-target effects. Nevertheless, AAV7 has been shown in a study to effectively transduce intestinal epithelial cells upon systemic delivery, making it one of the preferred serotypes for gut-targeted gene modulation (*Am J Physiol Gastrointest Liver Physiol.* 2012 Feb 1;302(3):G296-308.). To minimize the potential for off-target effects and ensure effective knockdown, we designed three different shRNA constructs targeting Hif-1 α and screened them in vitro prior to animal experimentation.

To evaluate their knockdown efficiency, we transduced Caco-2 cells, a human intestinal epithelial cell line, with individual AAV7-shHif-1 α constructs. After 72 hours, total RNA was extracted and subjected to RT-qPCR. The results showed that shRNA1, shRNA2, and shRNA3 reduced Hif-1 α mRNA expression by approximately 50%, 67%, and 80%, respectively, relative to the negative control group (Table 1). Based on these findings, shRNA3, which demonstrated the highest knockdown efficiency, was selected for subsequent in vivo experiments. Consistent with the in vitro findings, AAV7-shHIF1 α achieved efficient knockdown of HIF-1 α in vivo, as

confirmed by both RT-qPCR and Western blot analysis (Figure 1a-c). Overall, our results demonstrated that AAV7-shHif-1 α effectively and specifically silenced HIF-1 α expression, with no detectable off-target effects under the experimental conditions.

Table 1. Validation of shRNA Constructs Targeting Hif-1 α by RT-qPCR in Caco2 Cells

Well	Fluor	Target	Content	Sample	Biological Set Name	Cq	Cq Mean											
A04	SYBR	hif-1 α	Unkn	shRNA1		17.56	17.56											
A05	SYBR	hif-1 α	Unkn	shRNA1		17.68	17.68											
A06	SYBR	hif-1 α	Unkn	shRNA1		17.45	17.45											
B04	SYBR	hif-1 α	Unkn	shRNA2		17.87	17.87											
B05	SYBR	hif-1 α	Unkn	shRNA2		18.04	18.04											
B06	SYBR	hif-1 α	Unkn	shRNA2		18.02	18.02											
C04	SYBR	hif-1 α	Unkn	shRNA3		18.71	18.71											
C05	SYBR	hif-1 α	Unkn	shRNA3		18.71	18.71											
C06	SYBR	hif-1 α	Unkn	shRNA3		18.76	18.76											
D04	SYBR	hif-1 α	Unkn	NC-1		16.41	16.41											
D05	SYBR	hif-1 α	Unkn	NC-1		16.33	16.33											
D06	SYBR	hif-1 α	Unkn	NC-1		16.34	16.34											
A01	SYBR	Actin	Unkn	shRNA1		18.29	18.29											
A02	SYBR	Actin	Unkn	shRNA1		18.30	18.30											
A03	SYBR	Actin	Unkn	shRNA1		18.44	18.44											
B01	SYBR	Actin	Unkn	shRNA2		18.11	18.11											
B02	SYBR	Actin	Unkn	shRNA2		18.16	18.16											
B03	SYBR	Actin	Unkn	shRNA2		18.17	18.17											
C01	SYBR	Actin	Unkn	shRNA3		18.22	18.22											
C02	SYBR	Actin	Unkn	shRNA3		18.14	18.14											
C03	SYBR	Actin	Unkn	shRNA3		18.10	18.10											
D01	SYBR	Actin	Unkn	NC-1		18.20	18.20											
D02	SYBR	Actin	Unkn	NC-1		18.14	18.14											
D03	SYBR	Actin	Unkn	NC-1		18.10	18.10											

	Actin-CT	hif-1 α -CT	Δ CT		$\Delta\Delta$ CT	$2^{-\Delta\Delta$ CT}	$2^{-\Delta\Delta$ CT-mean
NC-1	18.20	16.41	-1.79		0.00	1.00	
NC-1	18.14	16.33	-1.81	-1.78436	-0.03	1.02	1.00
NC-1	18.10	16.34	-1.76		0.03	0.98	
shRNA1	18.29	17.56	-0.73		1.05	0.48	
shRNA1	18.30	17.68	-0.63		1.16	0.45	0.50
shRNA1	18.44	17.45	-0.99		0.80	0.57	
shRNA2	18.11	17.87	-0.24		1.54	0.34	
shRNA2	18.16	18.04	-0.12		1.67	0.31	0.33
shRNA2	18.17	18.02	-0.15		1.63	0.32	
shRNA3	18.22	18.71	0.48		2.27	0.21	
shRNA3	18.14	18.71	0.57		2.36	0.20	0.20
shRNA3	18.10	18.76	0.66		2.44	0.18	

Figure 1. PCR and Western blot confirmed successful knockdown of Hif-1 α .

Comment 6: In Fig. 6G, it is not clear which group received the AAV7 vector vs AAV7-Hif-1 α .

Response 6:

Thank you for your careful review. We apologize for the confusion caused by the labeling error during figure preparation. Additionally, due to the inclusion of supplementary experiments, the original Figure 6G has been changed to Figure 6i in the revised manuscript. The figure has been corrected to clearly indicate which groups received the AAV7 vector and which received the AAV7-Hif-1 α treatment.

Figure 6i

Comment 7: This sentence on “Nicotine exacerbates MASH severity via HIF-1 α knockdown in mice” is misleading. The data does not show that nicotine can knockdown HIF-1 α .

Response 7:

Thank you for the constructive review. We apologize for the misleading phrasing in the original statement, “Nicotine exacerbates MASH severity via HIF-1 α knockdown in mice”. This sentence may cause confusion. Our study focuses on investigating the effects of HIF-1 α knockdown on MASH in mice exposed to nicotine. To provide clearer expression, we have revised the sentence to (Lines: 446): “HIF-1 α knockdown exacerbates nicotine-exposed MASH in mice”.

Comment 8: The abstract mentioned “intestinal permeability facilitated bacterial translocation to the liver”. However, no such data was shown in the manuscript.

Response 8:

We appreciate the your comment. We apologize for the potentially misleading statement in the original abstract regarding bacterial translocation to the liver. Though our manuscript did not provide direct evidence of bacterial translocation, we did observe significantly increased serum lipopolysaccharide (LPS) levels in nicotine-exposed MASH mice (Figure 3j). Elevated serum LPS is widely accepted as an indirect indicator of enhanced gut permeability and bacterial translocation due to its role as a major component of Gram-negative bacterial cell walls. These findings support our interpretation that increased intestinal permeability may contribute to hepatic inflammation via

translocated bacterial products. However, to avoid ambiguity, we have deleted the relevant sentences in the abstract.

Figure 3j

Revised text reads:

Abstract: Nicotine accumulation in the intestine is associated with an exacerbation of metabolic dysfunction-associated steatohepatitis (MASH), but the underlying mechanisms remain enigmatic. We investigated how nicotine impacted intestinal microbiota composition and barrier function in MASH. Our study revealed significant intestinal microbiota dysbiosis and upregulated hypoxia-inducible factor 1-alpha (HIF-1 α) levels in nicotine-exposed MASH mice. HIF-1 α knockdown worsened intestinal barrier dysfunction in nicotine-exposed MASH mice. This exacerbation resulted from the suppression of MEK/ERK signaling pathway phosphorylation in HIF-1 α -deficient mice. Lactobacillus rhamnosus GG supernatant can alleviate hepatic injury in nicotine-exposed MASH mice; however, this protective effect was abolished in the absence of HIF-1 α . Taken together, this study reveals a critical pathogenic role of nicotine in exacerbating MASH through intestinal microbiota disruption and barrier dysfunction mediated by HIF-1 α . It also suggests exogenous probiotic supplementation as a potential therapeutic strategy for mitigating nicotine-induced MASH progression.

Comment 9: *It is not clear whether the control diet 1010086 is a purified diet, and whether it is isocaloric to CDAHFD. This need to be stated clearly. One criticism with the use of rodent chow (non-purified diet) is because it contains other plant-derived ingredients and fibers (i.e., not matched to CDAHFD) and therefore could confound the microbiota analyses. This is a minor*

comment because the main analyses compare CDAHFD to CDAHFD+nicotine, which is appropriate.

Response 9:

Thank you for this important comments. In our study, diet 1010086 was used as the standard low-fat control to assess MASH progression induced by the high-fat CDAHFD diet. Diet 1010086 is a purified diet containing 21.5% protein, 67.4% carbohydrates, and 11.1% fat, and has been widely employed in MASLD-related studies (*PMID: 39355870; PMID: 37676481; PMID: 40169574*). Importantly, all key mechanistic comparisons in this study were made between the CDAHFD and CDAHFD+nicotine groups, both fed identical diets, thus minimizing confounding variables related to diet composition. Therefore, the use of diet 1010086 as the control does not affect the validity of our conclusions.

Reviewer #3

Comment 1: *The major claims of the paper is that this study highlights the pivotal role of nicotine in exacerbating MASH through intestinal microbiota modulation and barrier dysfunction mediated via HIF-1 α regulation. This study is quite intriguing, as it scientifically explains the harmful effects of smoking on the human body, captures public attention, and guides a broad population towards adopting healthier habits. The findings are novel in linking nicotine-induced intestinal dysbiosis and HIF-1 α -mediated barrier dysfunction to MASH progression.*

Response 1:

Thank you for the positive and encouraging comments regarding our work.

Comment 2: *In the part of animal models, please develop a flowchart illustrating experimental animal grouping and feeding protocols to improve clarity and visual representation.*

Response 2:

Thank you for the insightful suggestion. We have added a detailed flowchart to Figure 1a in the revised manuscript.

Comment 3: *The manuscript is methodologically robust, employing a combination of in vivo models (CDAHFD diet + nicotine exposure), multi-omics approaches (16S rRNA sequencing,*

RNA-seq), and mechanistic studies (AAV-mediated HIF-1 α knockdown, MEK/ERK pathway inhibition). However, several technical limitations should be addressed:(1)While antibiotic (Abx) treatment effectively depletes microbiota, prolonged use may induce metabolic side effects (e.g., altered bile acid metabolism, immune modulation) that could confound results. The study does not account for these systemic effects, which may indirectly influence MASH progression.(2)The AAV7-HIF-1 α knockdown model lacks validation of tissue specificity (e.g., intestinal vs. hepatic HIF-1 α suppression). Off-target effects or incomplete knockdown could affect interpretation.

Response 3:

Thank you for the thoughtful comments. We have addressed both concerns as follows:

(1). In our study, we used a widely accepted pseudo-germ-free mouse model based on a combination of four antibiotics to evaluate the role of the intestinal microbiota in mediating nicotine's effects on MASH. This approach has been extensively applied in microbiome research (*Theranostics*. 2020 May 17;10(14):6500-6516; *Int J Mol Sci*. 2022 Aug 19;23(16):9350). We agree that Abx treatment may have systemic effects, such as alterations in bile acid metabolism or immune modulation, which could potentially influence the progression of MASH. We have now acknowledged this limitation in the revised Discussion section (Lines: 650-657).

Revised text reads: Another limitation of our study is the use of a pseudo-germ-free mouse model generated through long-term administration of a broad-spectrum antibiotic cocktail. Although this is a widely accepted and practical method for depleting the gut microbiota(14,34), it may also induce systemic metabolic effects, such as alterations in bile acid metabolism and immune function, which could potentially influence the progression of MASH. Antibiotic treatment may introduce confounding factors. Future studies using germ-free mice will be necessary to more accurately determine the causal role of the microbiota in mediating the effects of nicotine.

(2). AAV7 has been shown in a study to effectively transduce intestinal epithelial cells upon systemic delivery, making it one of the preferred serotypes for gut-targeted gene modulation (*Am J Physiol Gastrointest Liver Physiol*. 2012 Feb 1;302(3):G296-308.). We acknowledge that AAV7-mediated knockdown does not replicate the efficiency or intestinal specificity of a genetically engineered knockout model and may carry a risk of off-target effects. To minimize the

potential for off-target effects and ensure effective knockdown, we designed three different shRNA constructs targeting Hif-1 α and screened them in vitro prior to animal experimentation.

To evaluate their knockdown efficiency, we transduced Caco-2 cells, a human intestinal epithelial cell line, with individual AAV7-shHif-1 α constructs. After 72 hours, total RNA was extracted and subjected to RT-qPCR. The results showed that shRNA1, shRNA2, and shRNA3 reduced Hif-1 α mRNA expression by approximately 50%, 67%, and 80%, respectively, relative to the negative control group (Table 1). Based on these findings, shRNA3, which demonstrated the highest knockdown efficiency, was selected for subsequent in vivo experiments. Consistent with the in vitro findings, AAV7-shHIF1 α achieved efficient knockdown of HIF-1 α in vivo, as confirmed by both RT-qPCR and Western blot analysis (Figure 1a-c). Overall, our results demonstrated that AAV7-shHif-1 α effectively and specifically silenced HIF-1 α expression, with no detectable off-target effects under the experimental conditions.

Table 1. Validation of shRNA Constructs Targeting Hif-1 α by RT-qPCR in Caco2 Cells

Well	Fluor	Target	Content	Sample	Biologica 1 Set Name	Cq	Cq Mean											
A04	SYBR	hif.1a	Unkn	shRNA1		17.56	17.56											
A05	SYBR	hif.1a	Unkn	shRNA1		17.68	17.68											
A06	SYBR	hif.1a	Unkn	shRNA1		17.45	17.45											
B04	SYBR	hif.1a	Unkn	shRNA2		17.87	17.87											
B05	SYBR	hif.1a	Unkn	shRNA2		18.04	18.04											
B06	SYBR	hif.1a	Unkn	shRNA2		18.02	18.02											
C04	SYBR	hif.1a	Unkn	shRNA3		18.71	18.71											
C05	SYBR	hif.1a	Unkn	shRNA3		18.71	18.71											
C06	SYBR	hif.1a	Unkn	shRNA3		18.76	18.76											
D04	SYBR	hif.1a	Unkn	NC-1		16.41	16.41											
D05	SYBR	hif.1a	Unkn	NC-1		16.33	16.33											
D06	SYBR	hif.1a	Unkn	NC-1		16.34	16.34											
A01	SYBR	Actin	Unkn	shRNA1		18.29	18.29											
A02	SYBR	Actin	Unkn	shRNA1		18.30	18.30											
A03	SYBR	Actin	Unkn	shRNA1		18.44	18.44											
B01	SYBR	Actin	Unkn	shRNA2		18.11	18.11											
B02	SYBR	Actin	Unkn	shRNA2		18.16	18.16											
B03	SYBR	Actin	Unkn	shRNA2		18.17	18.17											
C01	SYBR	Actin	Unkn	shRNA3		18.22	18.22											
C02	SYBR	Actin	Unkn	shRNA3		18.14	18.14											
C03	SYBR	Actin	Unkn	shRNA3		18.10	18.10											
D01	SYBR	Actin	Unkn	NC-1		18.20	18.20											
D02	SYBR	Actin	Unkn	NC-1		18.14	18.14											
D03	SYBR	Actin	Unkn	NC-1		18.10	18.10											

	Actin-CT	hif.1a-CT	Δ CT	$\Delta\Delta$ CT	$2^{\Delta\Delta$ CT}	$2^{\Delta\Delta$ CT-mean
NC-1	18.20	16.41	-1.79		0.00	1.00
NC-1	18.14	16.33	-1.81	-1.78436	-0.03	1.02
NC-1	18.10	16.34	-1.76		0.03	0.98
shRNA1	18.29	17.56	-0.73		1.05	0.48
shRNA1	18.30	17.68	-0.63		1.16	0.45
shRNA1	18.44	17.45	-0.99		0.80	0.57
shRNA2	18.11	17.87	-0.24		1.54	0.34
shRNA2	18.16	18.04	-0.12		1.67	0.31
shRNA2	18.17	18.02	-0.15		1.63	0.32
shRNA3	18.22	18.71	0.48		2.27	0.21
shRNA3	18.14	18.71	0.57		2.36	0.20
shRNA3	18.10	18.76	0.66		2.44	0.18

Figure 1. PCR and Western blot confirmed successful knockdown of Hif-1 α .

Comment 4: *This work is highly relevant to hepatology, gastroenterology, and microbiome research. Key strengths include: Translational Implications: The gut-liver axis is a therapeutic hotspot for metabolic diseases. Demonstrating nicotine's role in disrupting this axis via HIF-1 α offers actionable targets (e.g., probiotics, HIF-1 α modulators). Public Health Impact: Smoking is a modifiable risk factor for MASLD/MASH. This study provides mechanistic evidence to support smoking cessation campaigns and probiotic interventions for smokers with liver disease.*

Response 4:

Thank you for the encouraging comments and for recognizing the translational and public health significance of our study.

Comment 5: *To enhance rigor and impact for the paper, if possible, please supplement HIF-1 α Rescue Experiments: Treat HIF-1 α -deficient mice with HIF-1 α agonists (e.g., FG-4592) to confirm reversibility of nicotine-induced damage.*

Response 5:

Thank you for this valuable suggestion. We agree that rescue experiments using HIF-1 α agonists, such as FG-4592, would further strengthen the mechanistic evidence by confirming the reversibility of nicotine-induced intestinal damage. However, this approach involves a separate pharmacological intervention that would substantially expand the scope and complexity of the current study. Future studies could evaluate whether pharmacological activation of HIF-1 α (e.g., FG-4592) can rescue intestinal barrier dysfunction and attenuate MASH in nicotine-exposed models.

Comment 6: *In terms of language, the introduction needs more in-depth description in order to attract readers' attention.*

Response 6:

Thank you for the constructive suggestion. In the revised manuscript, we have carefully revised

the Introduction to improve the language to better capture the reader's attention (Lines: 74-115).

Revised text reads:

Metabolic dysfunction-associated steatotic liver disease (MASLD) has emerged as the most common chronic liver disease worldwide, affecting approximately 30% of adults globally. Alarming, the global prevalence of MASLD is estimated to rise to 55% by 2040(1). MASLD, the fastest-growing cause of hepatocellular carcinoma, has an unclear pathogenesis. However, evidence links it to risk factors such as smoking, unhealthy diets, insulin resistance, type 2 diabetes mellitus, increased hepatic lipogenesis, and intestinal microbiota dysbiosis(2).

Smoking, as a well-recognized detrimental lifestyle habit, is known to increase the risk of lung and colorectal cancers(3). Nicotine, the primary addictive component of tobacco, has been identified as a key factor in smoking-related health issues(4). Our previous study demonstrated that nicotine induced mitochondrial dysfunction, oxidative stress, and apoptosis through suppression of CDGSH iron sulfur domain 3 expression, thereby exacerbating MASLD(5).

Intestinal dysbiosis and barrier dysfunction disrupt intestinal homeostasis, exacerbate systemic inflammation, and promote the translocation of harmful substances, further aggravating MASLD(6). Evidence indicates that nicotine accumulates in the intestine, exacerbating metabolic dysfunction-associated steatohepatitis (MASH). However, supplementation with intestinal microbiota has been shown to degrade intestinal nicotine and alleviate smoking-induced MASH(7). *Lactobacillus rhamnosus* GG is a widely studied probiotic that regulates the intestinal microbiota, improves lipid metabolism and inflammation, and effectively alleviates liver damage and metabolic disorders in high-fat diet-induced MASLD mice(8). Moreover, *Lactobacillus rhamnosus* GG enhances hypoxia-inducible factor 2 α signaling and upregulates the expression of intestinal tight junction proteins, thereby preserving intestinal barrier integrity and alleviating alcoholic liver injury(9). Hypoxia-inducible factor -1 α (HIF-1 α) plays a pivotal role in regulating intestinal homeostasis. It modulates intestinal epithelial integrity, and its dysregulation is associated with increased intestinal permeability and inflammatory signaling(10)(11). Our previous study demonstrated that mice with intestinal epithelial-specific knockout of HIF-1 α

exhibited more severe liver injury and higher serum lipopolysaccharides (LPS) levels in a murine model of alcoholic liver disease, further confirming the critical role of HIF-1 α in maintaining intestinal microbiota homeostasis and gut barrier integrity(10). However, the role of HIF-1 α in nicotine-exposed MASH remains inadequately understood.

In this study, we investigated how nicotine affects intestinal microbiota dysbiosis, barrier dysfunction, and HIF-1 α expression, elucidating the mechanisms by which nicotine exacerbates MASH. By providing new insights into the gut-liver axis, our findings enhance the understanding of smoking-related MASH pathogenesis and identify potential therapeutic targets.

Reviewer #2

Comment 1: *The term “pathogenic” should be changed to “pathologic” in the Abstract section. This is because nicotine is not a pathogen.*

Response 1:

Thank you for your valuable comment. We have revised the term “pathogenic” to “pathologic” in the Abstract accordingly.

Comment 2: *In the Abstract, the sentence on “...barrier dysfunction mediated by HIF-1 α ” is misleading, considering the author’s premise that HIF-1 α plays a protective role.*

Response 2:

Thank you for your insightful comment. To avoid the misleading implication that HIF-1 α mediates barrier dysfunction, we have revised the sentence in the Abstract. The phrase has now been changed to “which is associated with the downregulation of HIF-1 α in the intestine”.

Comment 3: *Please provide method and details on goblet cell staining.*

Response 3:

Thank you for your valuable suggestion. To address this point, we have added a detailed description of the AB-PAS staining procedure used to visualize goblet cells in paraffin-embedded ileal sections in the Materials and Methods section. **Revised text reads:** AB-PAS staining was performed on paraffin-embedded ileal sections to visualize goblet cells.

Comment 4: *RNA-seq detect changes in epithelial cells, but not the gut microbiota. The following sentence in Line 388 was incorrect. Line 388: “RNA sequencing analysis was performed on intestinal tissue samples to determine potential alterations in the intestinal microbiota of CDAHFD mice induced by nicotine exposure.”*

Response 4:

Thank you for your insightful comments. RNA-seq reflects transcriptional changes in intestinal epithelial tissue rather than alterations in the gut microbiota. Accordingly, the sentence has been revised to: “RNA sequencing analysis was performed on intestinal tissue samples to determine potential alterations in intestinal **gene** expression in CDAHFD mice induced by nicotine exposure”.

Reviewer #3

Comment 1: The authors need to provide western blot and qRT-PCR data of liver HIF-1 α (in the same figure where they had shown HIF-1 α expression in the intestine) after AAV7-HIF-1 α treatment.

Response 1: Thank you for your insightful comments. We have added the data as requested, including both qRT-PCR and western blot analyses of HIF-1 α in liver and intestinal tissues.

Revised text reads: As shown in Fig. 6f-h, AAV7-HIF-1 α markedly reduced HIF-1 α mRNA and protein levels in the ileum, whereas no significant change was observed in the liver compared with AAV7-Vector control. These results confirm that the AAV7 vector predominantly targets ileum tissue and achieves efficient HIF-1 α knockdown in the gut, while exerting minimal impact on hepatic HIF-1 α expression.

Fig. 6. HIF-1 α knockdown exacerbates nicotine-exposed MASH in mice. f-h, PCR and Western blot confirmed successful knockdown of Hif-1 α .